# Multimodal neuromarkers in schizophrenia via cognition-guided MRI fusion

Jing Sui [1,2,3,4], Shile Qi [1,3], Theo G.M. van Erp [5], Juan Bustillo[6], Rongtao Jiang[1,3], Dongdong Lin [2], Jessica A. Turner[2,7], Eswar Damaraju[2], Andrew R. Mayer[2,6], Yue Cui[1], Zening Fu[2], Yuhui Du [2], Jiayu Chen [2], Steven G. Potkin[5], Adrian Preda[5], Daniel H. Mathalon[8,9], Judith M. Ford[8,9], James Voyvodic[10], Bryon A. Mueller [11], Aysenil Belger[12], Sarah C. McEwen[13], Daniel S. O'Leary[14], Agnes McMahon [15], Tianzi Jiang [1,3,4] & Vince D. Calhoun [2,6,16]

Cognitive impairment is a feature of many psychiatric diseases, including schizophrenia. Here we aim to identify multimodal biomarkers for quantifying and predicting cognitive performance in individuals with schizophrenia and healthy controls. A supervised learning strategy is used to guide three-way multimodal magnetic resonance imaging (MRI) fusion in two independent cohorts including both healthy individuals and individuals with schizophrenia using multiple cognitive domain scores. Results highlight the salience network (gray matter, GM), corpus callosum (fractional anisotropy, FA), central executive and default-mode networks (fractional amplitude of low-frequency fluctuation, fALFF) as modality-specific biomarkers of generalized cognition. FALFF features are found to be more sensitive to cognitive domain differences, while the salience network in GM and corpus callosum in FA are highly consistent and predictive of multiple cognitive domains. These modality-specific brain regions define—in three separate cohorts—promising co-varying multimodal signatures that can be used as predictors of multi-domain cognition.

[1] Brainnetome Center and National Laboratory of Pattern Recognition, Institute of Automation, Chinese Academy of Sciences, 100190 Beijing, China. [2] The Mind Research Network, Albuquerque, NM 87106, USA. [3] University of Chinese Academy of Sciences, 100049 Beijing, China. [4] CAS Center for Excellence in Brain Science and Intelligence Technology, Institute of Automation, Chinese Academy of Sciences, 100190 Beijing, China. [5] Department of Psychiatry and Human Behavior, University of California, Irvine, Irvine, CA 92697, USA. [6] Department of Psychiatry, University of New Mexico, Albuquerque, NM 87131, USA. [7] Department of Psychology and Neuroscience, Georgia State University, Atlanta, GA 30302, USA. [8] Department of Psychiatry, University of California, San Francisco, CA 94143, USA. [9] San Francisco VA Medical Center, San Francisco, CA 94143, USA. [10] Department of Radiology, Brain Imaging and Analysis Center, Duke University, Durham, NC 27710, USA. [11] Department of Psychiatry, University of Minnesota, Minneapolis, MN 55454, USA. [12] Department of Psychiatry, University of North Carolina School of Medicine, Chapel Hill, NC 27599, USA. [13] Department of Psychiatry, University of California, San Diego, CA 92093, USA. [14] Department of Psychiatry, University of Iowa Carver College of Medicine, Iowa, IA 52242, USA. [15] USC Stevens Neuroimaging and Informatics Institute, University of Southern California, San Diego, CA 90033, USA. [16] Department of Electrical and Computer Engineering, University of New Mexico, Albuquerque, NM 87131, USA. These authors contributed equally: Jing Sui, Shile Qi. Correspondence and requests for materials should be addressed to J.S. (email: jing.sui@nlpr.ia.ac.cn) or to V.D.C. (email: vcalhoun@mrn.org)

Cognitive dysfunction is recognized as a core deficit in many psychotic disorders such as schizophrenia (SZ)[1]. Studies of psychopathology are now increasingly focusing on understanding how the human brain produces cognition, which may depend on knowledge of its large-scale organization. Mapping cognitive capability onto brain imaging[2] has the potential to provide clues by exploiting links among enriched types of imaging and behavioral information for individuals[3,4]. Here a neuromarker is defined as a brain measure that is associated with a cognitive or behavioral outcome that can predict individual performance[5]. Neuroimaging techniques like structural and functional magnetic resonance imaging (MRI) have enabled identification of neuromarkers that bring psychiatry from subjective descriptive classification into objective and tangible brain-based measures[6,7]. For example, Rosenberg et al.[8] demonstrated that whole-brain functional connectivity strength may serve as a neuromarker of sustained attention for both healthy and disease assessments. Finn et al.[9] showed that functional connectivity (FC) profiles can predict levels of fluid intelligence. Drysdale et al.[10] further used FC patterns to define novel subtypes of depression that may benefit treatment outcome prediction. These studies provided steps in developing biomarkers that will allow the field of imaging analysis and psychiatry to move forward to a new era, which motivates our work in identifying neuromarkers that are associated with cognitive composite ability[11,12] and specific cognitive domains such as attention, working memory, and verbal learning.

The identified neuromarkers are most meaningful when they are replicable and can be used to predict new, previously unseen subjects. Though receiving increased attention[13,14], current neurocognitive investigations are often focusing on one cognitive domain or analyzing one single modality, or performing correlations after separate unimodal analyses[15]. Therefore, the multi-domain and multimodal cross-information are either missing or not being fully leveraged to improve neuromarker identification, despite the evidence that such information is highly informative[16–18]. To address the above-mentioned issues, we performed successive studies in this paper to answer the following four questions using three independent subject cohorts ($n = 294$, $n = 83$, and $n = 88$), which both have three types of MRI data and are measured with similar cognitive metrics, respectively.

First, what combination of multimodal brain networks will be associated with the global cognitive ability, especially when impaired in schizophrenia? Can the identified multimodal neuromarker signatures be replicated in another independent dataset? Secondly, how are the identified neuromarker signatures associated with other cognitive domains? What are the most correlated cognitive domains? Furthermore, which brain networks are associated with specific representative cognitive domains? What are their commonality and differences across domains? Which imaging modality is more sensitive to cognitive domain discrepancy? Finally, are the identified multimodal brain networks able to predict cognitive performance for new individuals?

To this end, we searched for multimodal neuromarker signatures that can be used to quantify and predict cognitive performance, especially impaired in schizophrenia, by successive multivariate data mining and model generalization. According to the triple network model of major psychopathology proposed by Menon[19], the aberrant intrinsic organization and interaction of the salience network (SAN), central executive network (CEN), and default-mode network (DMN) is characteristic of many psychiatric and neurological disorders. Considering both structural and functional dysfunction in these three networks have been found in schizophrenia[20–22] and linked with cognitive deficits[23,24], we hypothesize that the modality-specific SAN, CEN, and DMN would have pivotal roles in cognitive impairment in schizophrenia[23], which may consist multimodal signatures that are defined as modality-specific brain regions[25] exhibiting similar subject-wise covariation that can jointly predict cognitive performance for unseen individuals.

In order to address the first three questions, cognitive composite and multiple domain scores were used as references, respectively, to guide the three-way multimodal MRI data fusion in a discovery cohort: FBIRN (Function Biomedical Informatics Research Network, $n = 294$) by our proposed model[26] (see Method for the analysis flowchart). Results were further replicated in an independent cohort ($n = 83$). After discovering the multimodal signatures particularly associated with each of the four cognitive domains (including Computerized Multiphasic Interactive Neuro-cognitive System [CMINDS][27] composite, CMINDS attention, CMINDS working memory, and CMINDS verbal learning), we compared them in two ways: firstly, across domains—to reveal the commonality and uniqueness of the domain-specific multimodal signatures; secondly, across modalities—to evaluate which imaging modality is more sensitive to cognitive domain discrepancy. Finally, by taking advantage of the above-extracted brain networks in one cohort, we built predictive models of cognitive metrics by performing linear regression on these identified neuromarkers. Remarkably, these models were successful in predicting the corresponding cognitive metrics for new individuals in another two independent cohorts: UNM (University of New Mexico, $n = 83$) and COBRE (Center for Biomedical Research Excellence, $n = 88$)[28] This validates the generalizability of the identified modality-specific potential neuromarkers, which might be broadly applicable as predictors of multi-domain cognitive scores for new individuals.

## Results

**Multimodal networks associated with cognitive composite scores.** We aim to identify multimodal co-varying and modality-specific brain networks associated with composite cognitive scores. Methods section presents the whole-analysis flowchart of data fusion and prediction. First, both the CMINDS composite score and the MCCB composite score were used as the reference for FBIRN and UNM cohorts, respectively, with MCCAR + jICA[26,29] (multimodal canonical correlation analysis with reference plus joint independent component analysis). Then, to test the replicability of the identified multimodal networks, we calculated the permuted spatial correlations of their maps between cohorts and summarized the most affected cognitive domains related to schizophrenic deficit. Furthermore, cognitive domain scores of CMINDS attention, CMINDS working memory, and CMINDS verbal learning were used as reference to guide the three-way MRI fusion, respectively, aiming to identify the domain-common and modality-specific neuromarkers. Finally, after extracting the neuromarker features across multiple cognitive domains, we built linear regression models to predict individualized cognitive scores in FBIRN cohort, which were further generalized to predict corresponding cognitive measures of unseen subjects in two independent cohorts (UNM and COBRE).

The FBIRN Phase III study consisting of 147 SZs (mean $39.5 \pm$ std11.7) and 147 HCs ($37.4 \pm 11$), who were each measured by a neuropsychological battery that includes six neurocognitive domain tests called CMINDS[27].

Three representative MRI features (fractional amplitude of low-frequency fluctuations (fALFF) from resting-state functional MRI (rs-fMRI), gray matter (GM) density from structural MRI (sMRI), and fractional anisotropy (FA) from diffusion MRI (dMRI)) were combined by a fusion with reference model[26], in which CMINDS composite cognitive scores were adopted as the

reference. MCCAR + jICA[26] can simultaneously maximize the inter-modality covariation and correlations of certain imaging components with the referred clinical or cognitive measures (see more details in the Methods section). Based on the minimum description length (MDL)[30] criterion, 24 components were estimated. After joint decomposition, independent components (ICs) and their subject-wise loadings were derived for each modality.

The supervised fusion model produced one joint IC showing significant correlations with the CMINDS cognitive composite scores for all modalities, and we denote this IC as $IC_{ref}$ (Fig. 1). Figure 1a displays the spatial maps of $FBIRN\_IC_{ref\_composite}$ for each modality. Figure 1b indicates that $FBIRN\_IC_{ref\_composite}$ has positive correlations with the CMINDS composite scores for all modalities ($r = 0.486^*, 0.262^*, 0.430^*$ for sMRI, dMRI, and fMRI, respectively, * means FDR corrected for multiple comparisons, which represents the same meaning for all correlations and $p$ values in this paper), namely, the higher the loadings, the better the cognitive performance. The $p$ values derived from a permutation test for the correlation (details provided in Supplementary Note 1 and Supplementary Fig. 1) between $FBIRN\_IC_{ref\_composite}$ and composite cognitive scores are $p_{perm} = 1.3 \times 10^{-4*}, 0.002, 1.0 \times 10^{-4*}$ for sMRI, dMRI, and fMRI,

respectively (Fig. 1b). Moreover, the FA and GM components also show significant correlations with Positive and Negative Syndrome Scale (PANSS)[31] negative scores (dMRI: $r = -0.162$, $p = 0.05$, sMRI: $r = -0.285^*$, $p = 5.3 \times 10^{-4}$), which share many features with cognitive impairment[32,33]. No significant correlations were found with PANSS positive scores. The identified brain regions in $FBIRN\_IC_{ref\_composite}$ are summarized in Supplementary Table 1 for fALFF components (Talairach labels), FA (WM tracts, from John Hopkins Atlas), and GM (Montreal Neurological Institute labels) respectively. Two sample $t$-tests were also performed on IC loadings between patients and controls. Notably, the $FBIRN\_IC_{ref\_composite}$ also differs significantly in each feature with lower means in SZ ($p = 7.5 \times 10^{-17*}, 2.7 \times 10^{-7*}, 5.3 \times 10^{-12*}$), as shown in Fig. 1c, indicating a co-varying fALFF-FA-GM decrease in SZ. We denote this $FBIRN\_IC_{ref\_composite}$ as a joint multimodal signature that is closely associated with the cognitive composite scores.

**Cross-cohort replication.** Due to the interferential effects of varying demographic distributions and measurement conditions, few neurocognitive studies have been strictly replicated for independent cohorts. Here, to validate the replicability of the above-extracted multimodal networks related to composite

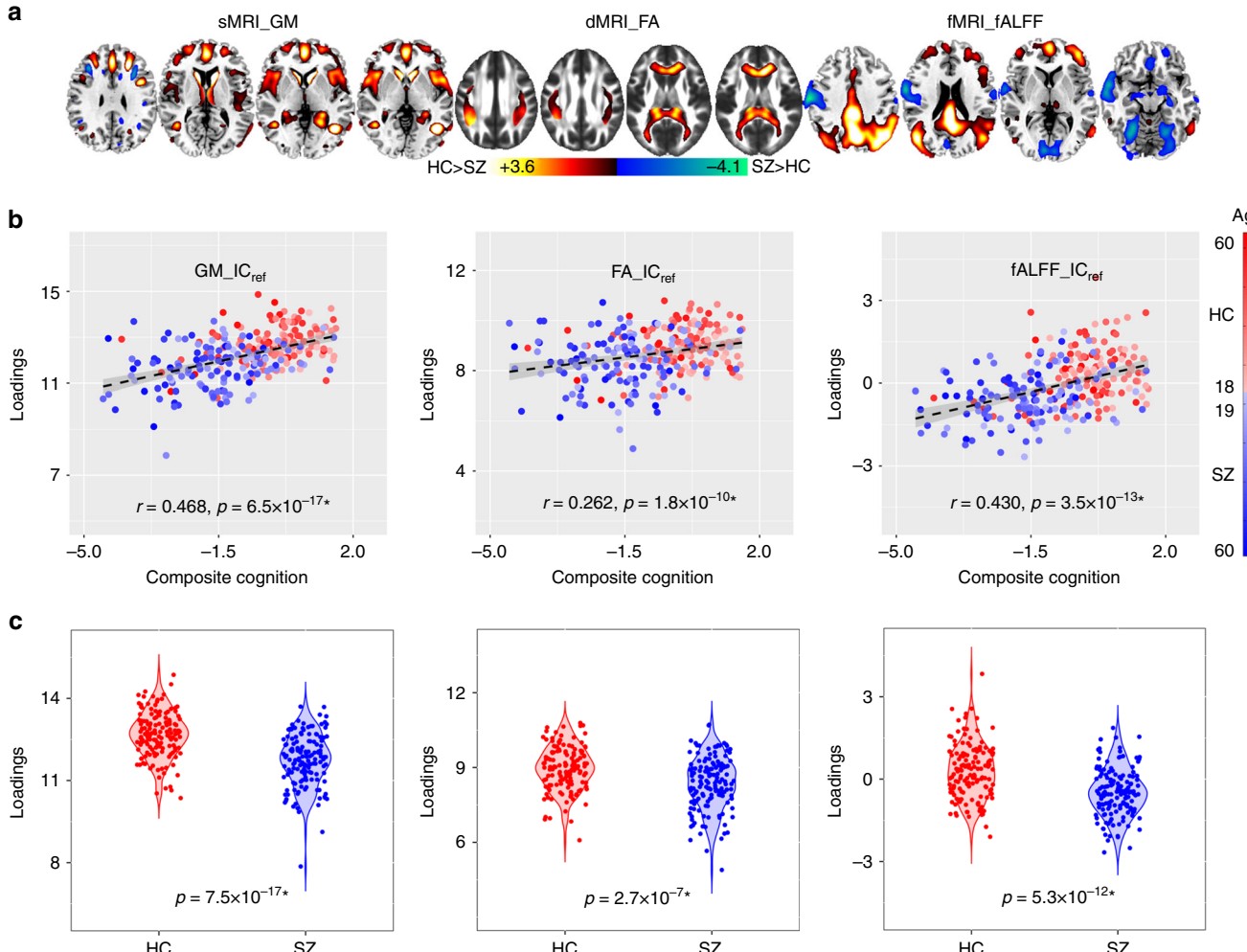

**Fig. 1** The identified joint components for FBIRN cohort. **a** The brain maps visualized at |Z| > 2; the positive values (red) means SZ < HC and the negative values (blue) means SZ > HC. **b** Correlations between CMINDS composite scores and loadings of component (HC: the red dots, SZ: the blue dots); thus SZ corresponds to worse cognitive performance and lower loading weights compared to HC. **c** Boxplot of the loading parameters of $FBIRN\_IC_{ref\_composite}$ that were adjusted as HC > SZ on the mean of loadings for each modality, with two sample $t$-tests $p$ values between HC and SZ shown bottom. The gray regions in **b** indicate a 95% confidence interval

cognitive scores, we analyzed an independent cohort including 44 HCs and 39 SZs collected from UNM, which incorporated a different but related cognitive battery called the MATRICS (Measurement and Treatment Research to Improve Cognition in Schizophrenia) Consensus Cognitive Battery (MCCB)[34]. There was no subject overlap between the UNM and FBIRN and COBRE data. MCCB composite was used as reference to guide the 3-way fusion of UNM data. Twenty components were estimated based on MDL criterion[30]. A similar joint $IC_{ref}$ (denoted as UNM_ $IC_{ref\_composite}$) was identified to be associated with the MCCB composite scores ($r = 0.271^*, 0.281^*, 0.311^*$) and also group-discriminating ($p = 0.005^*, 0.043, 0.002^*$ for sMRI, dMRI, and fMRI, respectively), as displayed in Fig. 2. The identified areas in UNM_ $IC_{ref\_composite}$ are summarized in Supplementary Table 2. The $p$ values from the permutation test for the correlation between UNM_$IC_{ref\_composite}$ and composite cognitive scores are 0.02, 0.01, 0.001 for sMRI, dMRI, and fMRI, respectively (Fig. 2b).

Notably, the identified multimodal networks exhibit substantial spatial overlap between discovery cohort (FBIRN) and replication cohort (UNM), while the cognitive performance was evaluated from two different systems (CMINDS and MCCB), which are similar but not identical in domain tasks[27]. Figure 3 summarizes

the similarity of global cognition-associated brain maps between independent cohorts and modalities. The cross-cohort correlation between the identified components are GM: $r = 0.51$, $p_{perm} = 8.0 \times 10^{-4}$, FA: $r = 0.59$, $p_{perm} = 2.0 \times 10^{\alpha}$; fALFF: $r = 0.45$, $p_{perm} = 0.002$, where the significance $p_{perm}$ were resulted from 10000 permutations. Details provided in Supplementary Note 2 and Supplementary Table 3.

As shown in Fig. 3, sMRI and dMRI features show high spatial similarity ($r > 0.5$) between cohorts. Particularly, decreased GM volume in the salience network (SAN, including dorsal anterior cingulate cortex [ACC] and insula), dorsolateral prefrontal cortex (dlPFC, key node of CEN), and subcortical clusters (including striatum and thalamus) were detected in SZ for both cohorts. For dMRI, the FA reduction in SZ on white matter tracts (such as corpus callosum[35], superior longitudinal fasciculus [SLF][36] and anterior thalamic radiation [ATR][37]) were common for both cohorts. For fMRI, the prefrontal regions and posterior DMN were more consistent between cohorts with a lower fALFF in SZ.

Therefore, three key brain networks (SAN, CEN, pDMN) were identified in different modalities to be closely related to cognitive performance, which are also replicated in two independent cohorts (Figs. 1–3 and Supplementary Tables 1, 2). Note that these three networks have previously been associated with

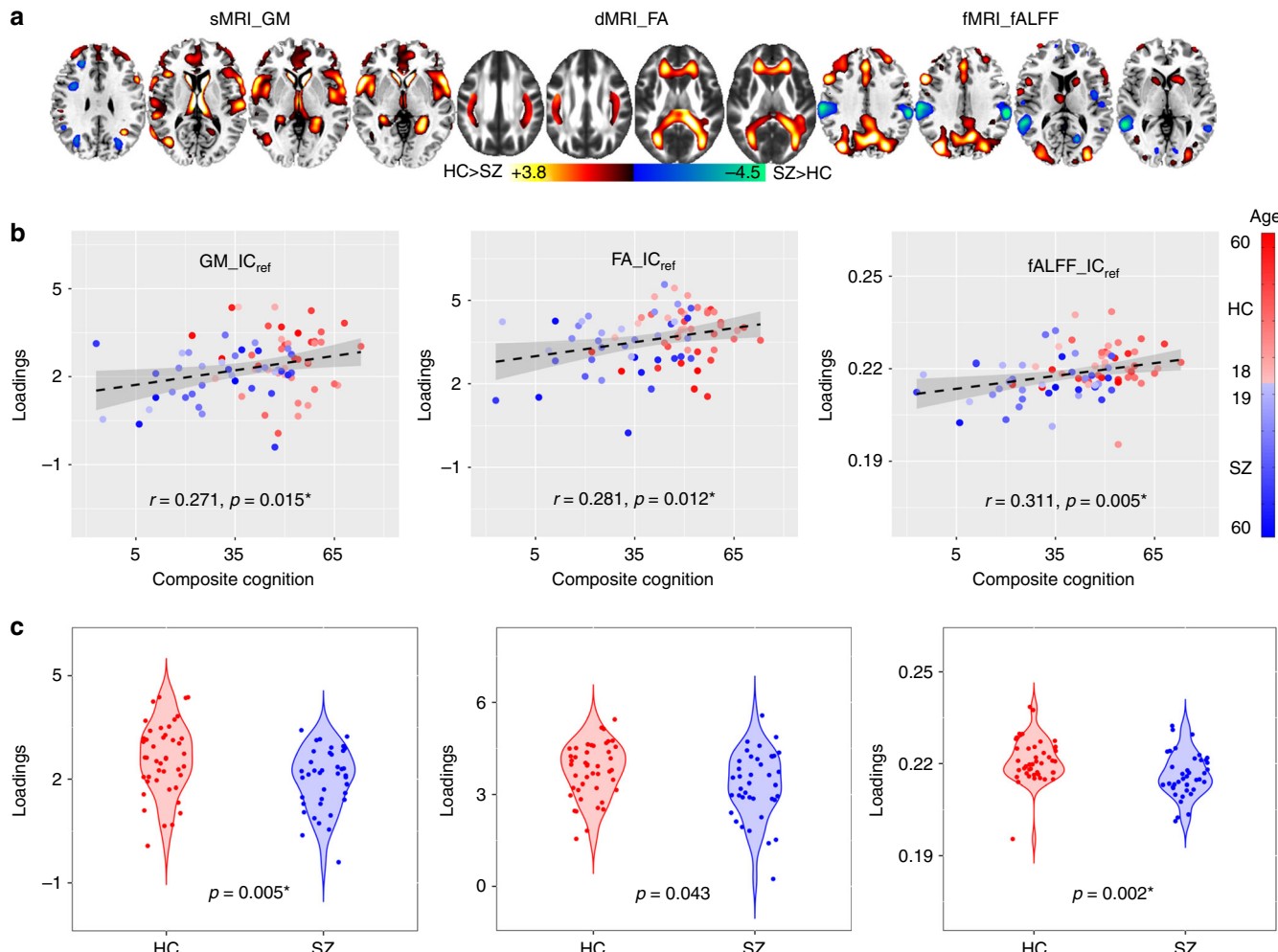

**Fig. 2** The identified joint components for UNM cohort. **a** The brain maps visualized at $|Z| > 2$; the positive values (red) means SZ < HC and the negative values (blue) means SZ > HC. **b** Correlations between loadings of $IC_{ref}$ and MCCB composite scores (HC: the red dots, SZ: the blue dots); thus SZ corresponds to worse cognitive performance and lower loading weights compared to HC. **c** Boxplot of the loading parameters of UNM_$IC_{ref\_composite}$ that were adjusted as HC > SZ on the mean of loadings for each modality, with two sample $t$-tests $p$ values between HC and SZ as shown at the bottom. The gray regions in **b** indicate a 95% confidence interval

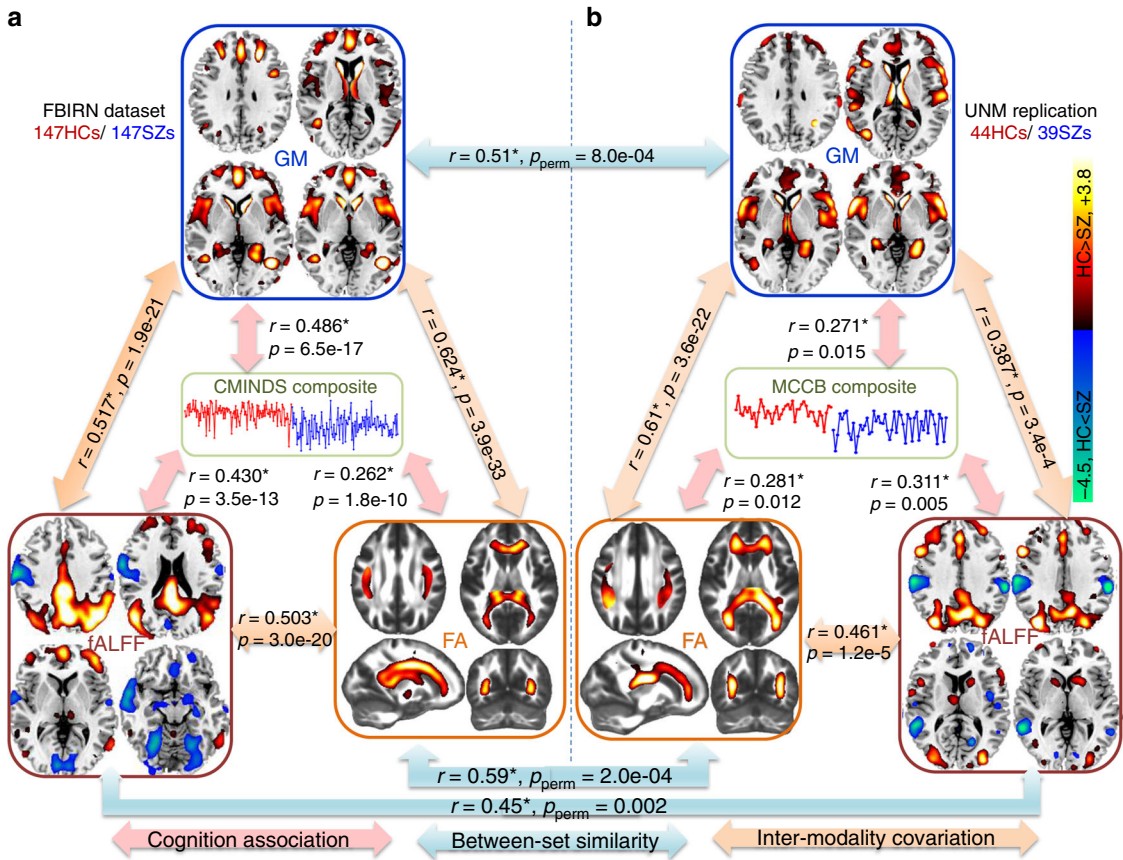

**Fig. 3** Similarity and co-varying patterns of the identified multimodal networks between cohorts. **a** The identified multimodal networks associated with the CMINDS composite score in the FBIRN cohort. **b** The identified multimodal networks associated with the MCCB composite score in the UNM cohort. The brain maps were visualized at $|Z| > 2$. The blue arrows represent the cross-cohort spatial correlation ($p_{perm}$ are $p$ values from a permutation test using 10,000 randomizations); the orange arrows represent inter-modality covariation across subjects; and the pink arrows denote the correlations between cognitive composite score (the reference) and the identified $IC_{ref\_composite}$

cognitive deficits in SZ in multiple reports[38–41], but never in a multimodal analysis across multiple cohorts and domains. More importantly, the three networks are in accord with our initial hypothesis based on the triple network theory[19] that aberrant organization and functioning within CEN, SAN, and DMN is closely associated with cognitive impairment in psychiatric disorders. Next, we investigated changes in the three networks in SZ in detail, as well as the associations of the identified multimodal components with other cognitive domains and different modalities.

**Association with other cognitive domains**. Cognitive function encompasses a variety of cognitive domains, including but not limited to attention, working memory, visual/verbal learning, decision-making, reasoning, and social cognition[12]. We next investigate the association between the identified multimodal components with different cognitive domains by calculating loadings of FBIRN_$IC_{ref\_composite}$ in each modality with six CMINDS domain scores (Supplementary Table 4). Results indicate that the top three most correlated domains are attention ($r = 0.441^*, 0.269^*, 0.333^*$), working memory ($r = 0.402^*, 0.240^*, 0.341^*$), and verbal learning ($r = 0.427^*, 0.232^*, 0.361^*$) for sMRI, dMRI, and fMRI, respectively, as displayed at the bottom of Fig. 4.

Figure 4 summarizes the key neurocognitive networks identified in FBIRN_$IC_{ref\_composite}$ and their associated cognitive domains with different colors. Specifically, there are higher fALFF values in pDMN (blue), including the ventromedial

prefrontal cortex (vmPFC) and posterior cingulate cortex (PCC), but lower fALFF in the middle temporal gyrus (MTG) and primary visual cortex (V1) associated with a higher cognitive composite score[42] (see also Fig. 1). Additional key compartments of CEN[43] (purple), including dlPFC, superior temporal gyrus (STG), and posterior parietal cortex (PPC), were also identified in fALFF. For GM, higher GM volume in the salience network[44] (cyan; ACC and bilateral insula) and subcortical regions (red; including hippocampus, caudate, and thalamus) are correlated with higher cognition, especially in the domain of attention. For dMRI, SZ shows lower FA values than HC in major white matter tracts (orange arrows) including forceps major (FMAJ), forceps minor (FMIN)[35], SLF[36], and ATR[37], which can potentially connect the identified brain regions in GM or fALFF. For instance, the FMIN (genu of the corpus callosum) interconnect the left and right frontal lobe, linking dlPFC and vmPFC (both were detected in fALFF and GM), and provide evidence that disrupted anatomical connections in SZ in the anterior corpus callosum may relate with the PFC impairment in both function and structure, as well as cognitive deficits in multiple domains such as working memory and verbal learning[45]. Similarly, the identified regions in CEN in fALFF are also connected by the SLF, which has been implicated in poor executive function in SZ[35]. These results clearly suggest that alterations in one modality can be associated with correlated changes in distant, but connected regions in another modality (Figs. 3 and 4) and multimodal fusion proves to be a powerful tool to reveal this association.

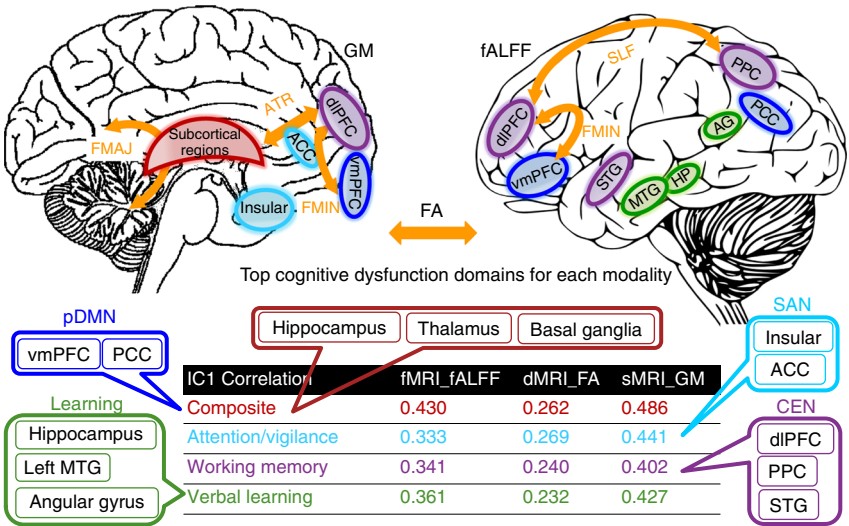

**Fig. 4** Key neurocognitive networks and the correspondence with CMINDS cognitive domains. Loadings of the identified component (FBIRN_IC$_{ref\_composite}$) are correlated with the cognitive composite score and multiple domain scores for each modality. Here, we listed the top three most correlated domains. The CMINDS domains are shown in the same color with its most correspondent brain networks; consequently, the brain regions in the top panel are demonstrated with the same color of its corresponding network. Left panel is GM, right panel is fALFF, FA is illustrated as orange arrows, MTG is middle temporal gyrus, PPC is posterior parietal cortex, PCC is posterior cingulate cortex

**Multimodal brain networks associated with three particular cognitive domains**. Following the above findings, attention, working memory, and verbal learning are the top three domains that are most correlated with the multimodal networks associated with global cognition and all are impaired in SZ. Next, we investigated the multimodal brain networks that are specifically associated with these three domains, respectively; which were then further compared in two ways: firstly, across domains to reveal the commonality and uniqueness of the multimodal brain regions associated with particular cognitive domains and secondly, across modalities to evaluate which imaging modality is more sensitive to cognitive domain discrepancy, as illustrated in Fig. 5.

Toward this goal, CMINDS domain scores of working memory, attention or verbal learning were used as reference, respectively, to perform the same reference-directed fusion analysis as we did using CMINDS composite scores (see Methods section). For each cognitive domain, we extracted one joint IC$_{ref}$ that is significantly correlated with specific domain scores and also group-discriminative in all modalities (all passed FDR correction at $p < 0.05$; see Supplementary Table 5). In order to quantify the robustness of the cognition-brain correlation across groups, we performed partial correlation to minimize the group effect. As shown in Supplementary Table 6, in the case of partial correlation, the cognition-imaging correlations remain significant (FDR corrected) after controlling for diagnosis in all four domains.

Figure 5a–d displays spatial maps of the identified FBIRN_IC$_{ref\_composite}$ (red), FBIRN_IC$_{ref\_memory}$ (green), FBIRN_IC$_{ref\_attention}$ (cyan), and FBIRN_IC$_{ref\_learning}$ (magenta), respectively. Figure 5e indicates the pairwise cross-domain similarity calculated by performing 3D correlations among the spatial maps of FBIRN_IC$_{ref\_composite}$, FBIRN_IC$_{ref\_memory}$, FBIRN_IC$_{ref\_attention}$, and FBIRN_IC$_{ref\_learning}$ for GM, FA, and fALFF, respectively. Note that the darker blue and larger shading denotes a higher correlation. GM and FA maps demonstrate more consistent patterns across cognitive domains ($r > 0.6$), while fALFF maps exhibit more variance ($r < 0.5$ but still significant) and are more sensitive to domain differences. In particular, our results revealed a pattern of overlapping GM reduction in SAN and dlPFC across all domains, which is consistent with most recent meta-analyses

in SZ patients. Hippocampus volume reduction is associated with the composite, working memory, and verbal learning domains. SZ patients also show decreased FA values in FMAJ, FMIN and ATR for all domains. For fALFF, the prefrontal cortex is consistently observed in all domains, while subcortical regions are more variable across domains.

In order to examine the domain-specific fALFF patterns more precisely, we display the representative slices of FBIRN_IC$_{ref\_composite}$, FBIRN_IC$_{ref\_memory}$, FBIRN_IC$_{ref\_attention}$, and FBIRN_IC$_{ref\_learning}$. Figure 6a shows both positive and negative fALFF patterns. Figure 6b plots correlations between IC$_{ref}$ loadings with its corresponding domain scores after controlling for diagnosis (Supplementary Table 6). It is clear that vmPFC and dlPFC are the domain-common fALFF regions that have been widely implicated in higher-order cognitive functions (e.g., attention, working memory, and verbal learning), as reflected in our results. Lower fALFF values in SZ were observed, corresponding to poorer cognitive performance than HCs[46,47], which is also consistent with two related FBIRN fMRI studies[48]. In addition, pDMN was identified only for the cognitive composite and working memory domains, while other subcortical and cortical regions such as thalamus, hippocampus, STG, and visual cortex occur differently depending on the different domains. For instance, the increased fALFF in hippocampus in SZ is only detected for the verbal learning and composite scores, which is consistent with a recent result that suggests low-frequency hippocampal–cortical activity drives brain-wide resting-state fMRI connectivity and contributes to cognition[49].

Our findings suggest that functional measures such as fALFF is more sensitive to differences between cognitive domains, whereas structural brain measures (GM/FA) are more similar and replicable across multi-domain cognitive impairment. Furthermore, in order to characterize the common core brain networks across the four cognitive domains (composite, working memory, attention, verbal learning), we extracted the overlapped brain regions of FBIRN_IC$_{ref\_composite}$, FBIRN_IC$_{ref\_memory}$, FBIRN_IC$_{ref\_attention}$, and FBIRN_IC$_{ref\_learning}$ (see details in the Methods section on neuromarker extraction) and summarized the domain-common brain patterns for each modality as shown in Fig. 7a. Evidently, the SAN in GM (red), corpus callosum (CC) in FA

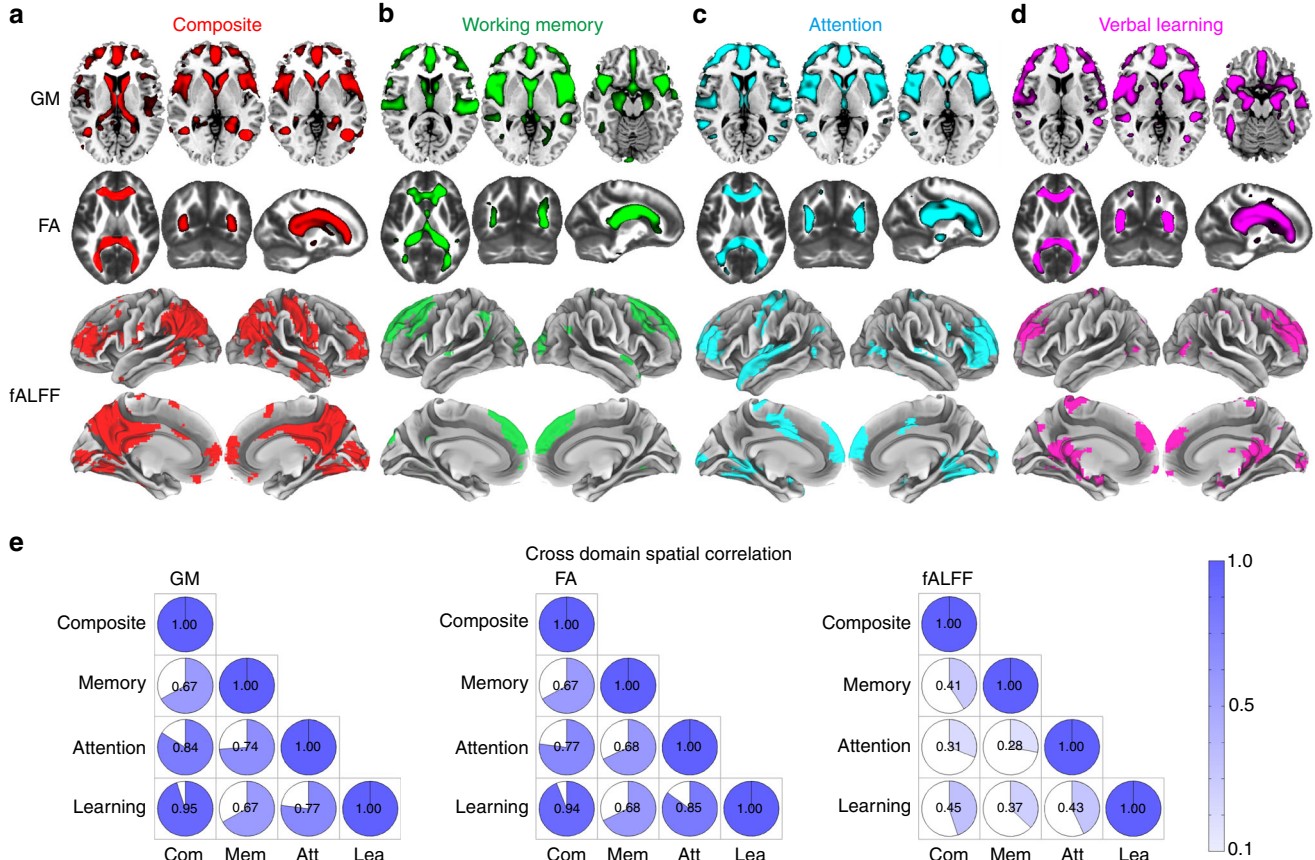

**Fig. 5** Comparison of multimodal components associated with the four specific cognitive domains. The spatial maps of the neuromarker network obtained under the guidance of reference **a** composite score (red), **b** working memory (green), **c** attention (cyan) and **d** verbal learning (magenta) are displayed in different color mapping. **e** The pairwise, cross-domain, spatial correlation among brain maps of FBIRN_IC_ref_composite, FBIRN_IC_ref_memory, FBIRN_IC_ref_attention, and FBIRN_IC_ref_learning for GM, FA and fALFF respectively. The darker blue and larger shading denotes higher correlation. Note that GM and FA maps demonstrate more consistent patterns across cognitive domains ($r > 0.6$), while fALFF maps exhibit more variance ($r < 0.5$ but still significant). This suggests that a functional measure such as fALFF may differentiate between cognitive domains more sensitively, whereas structural brain patterns (GM/FA) are more consistent across cognitive domains

(green), and PFC (dlPFC + vmPFC) in fALFF (yellow) are shared among all four domains, which may be treated as potential neuromarkers in each modality to characterize cognition quantitatively.

Based on the masks displayed in Fig. 7a, we further extracted the corresponding regions of interest (ROIs) in each modality to examine their original group differences and association with cognition (before fusion). The mean voxel values of each ROI were calculated for each subject and each modality. As expected, two sample $t$-tests between HC and SZ show significant group differences for GM_SAN ($p = 6.8 \times 10^{-9*}$), FA_CC ($p = 0.001$), and fALFF_PFC ($p = 7.0 \times 10^{-9*}$), respectively. The effect size for separating groups was also computed with Cohen's $d = 0.698$, $0.487$, and $0.623$ for GM_SAN, FA_CC, and fALFF_PFC, respectively. In addition, the GM_SAN, FA_CC, and fALFF_PFC also show significant correlations with all CMINDS cognitive domain scores, as listed in Supplementary Table 7. All above results suggest that these extracted ROIs may have pivotal roles as neurocognitive substrates, i.e., so-called neuromarker signatures, as they are common core brain networks among multiple cognitive domains, and are widely impaired in mental disorders[50].

**Predicting individual cognitive performance**. An ultimate goal of using neuromarkers for neuro-prognosis is to perform individualized predictions of educational or health outcomes[5]. To

verify the predictability on individual cognitive performance of the identified neuromarker signatures, we used the above-extracted domain-common ROIs (GM_SAN, FA_CC, and fALFF_PFC) plus fALFF_pDMN to predict the CMINDS composite scores in the FBIRN cohort. Here, fALFF_pDMN is included due to the triple network hypothesis[19] and its occurrence in two domains (composite and working memory). The mean voxel values of each ROI (neuromarker) were then calculated as regressors, which were used to conduct a multiple linear regression for the CMINDS composite score, achieving Eq. (1) (also see details in the Methods section on multiple linear regression):

$$\begin{aligned} \text{CMINDS composite} = &-0.8 + \text{GM\_SAN} \times 0.34 \\ &+ \text{FA\_CC} \times 0.19 + \text{fALFF\_PFC} \times 0.12 \\ &+ \text{ALFF\_pDMN} \times 0.13 \end{aligned} \quad (1)$$

Based on Eq. (1), correlation of $r = 0.463^*$ was achieved between the estimated CMINDS composite scores and its true values (Fig. 7b).

**Generalized prediction to independent cohorts**. To test the generalizability of the identified four neuromarkers and the prediction model, we then extracted the same four ROIs in UNM and COBRE cohorts through masks (obtained from FBIRN cohort) and applied them to Eq. (1), to predict the unseen MCCB composite

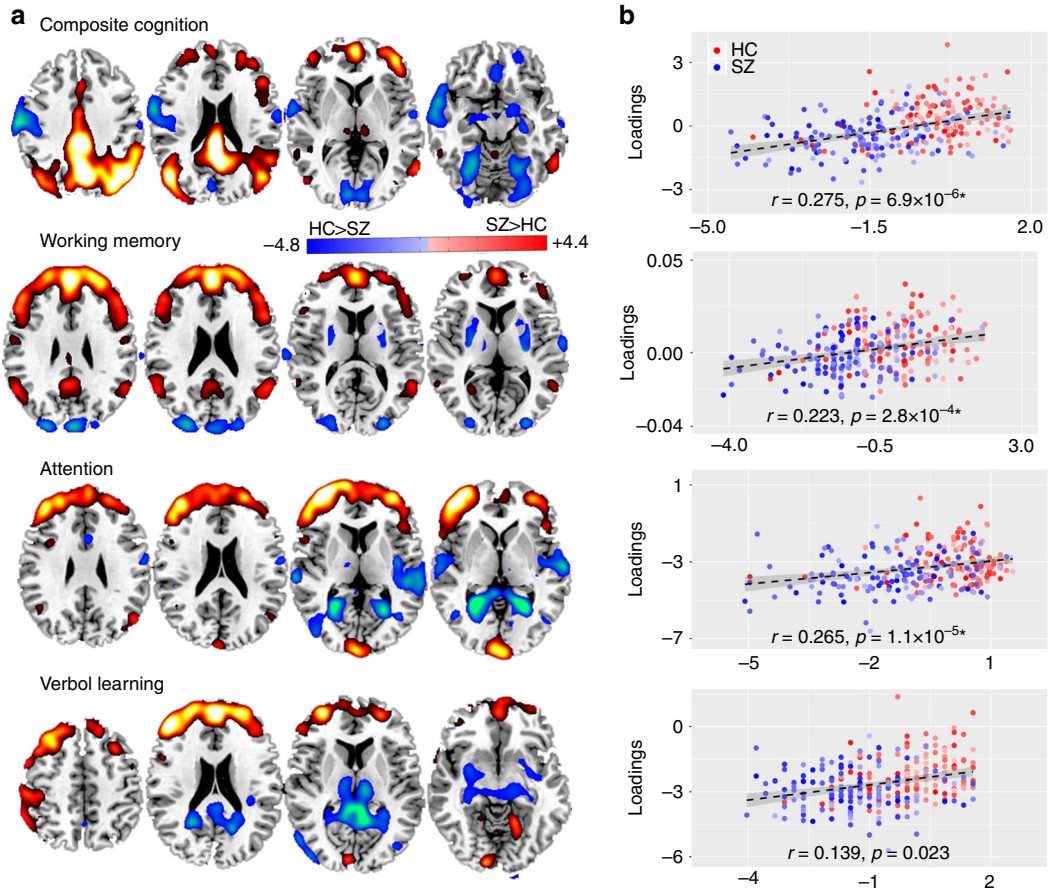

**Fig. 6** Spatial maps of fALFF activations for four main CMINDS cognitive domains. **a** The brain maps of fALFF FBIRN_IC$_{ref}$ associated with composite, working memory, attention and verbal learning domains respectively that are visualized at $|Z| > 2$. The positive values (red) means SZ < HC and the negative values (blue) means SZ > HC. **b** Correlations between loadings of fALFF component and its associated CMINDS domain scores after controlling for diagnosis (HC: the red dots, SZ: the blue dots); the higher loadings correspond to better cognitive performance. The gray regions in **b** indicate a 95% confidence interval

scores for both UNM and COBRE data (preprocessed using the same pipeline as in FBIRN). The model generalization from CMINDS to MCCB work well not only for composite score, but also for most cognitive domains (see Table 1 below), implicating the appreciable generalizability of the cognition-predictive models. CMINDS and MCCB are two similar but not identical cognitive measurement systems[27]; therefore, the cross-cohort generalization is a powerful evidence to validate the predictability of global cognition using the identified neuromarkers and model. As shown in Fig. 7d and c, Pearson correlations of $r = 0.406$ and $r = 0.236$ were achieved between the estimated MCCB composite scores and its true values for COBRE (42 HCs/46SZs) and UNM (41HCs/37SZs), respectively, suggesting good generalizability of the proposed cognition-prediction model. Note that the prediction models in Fig. 7b–d are the same, i.e., training in the FBIRN cohort to predict the CMINDS composite. Moreover, we also performed the prediction analysis within each group (HC or SZ) based on the four neuromarker signatures, i.e., using the group model trained by FBIRN to predict UNM + COBRE. The generalization in either case works well as shown in Supplementary Fig. 2 and Supplementary Note 3.

**Predictability on multiple cognitive domains**. In addition to global cognitive performance, we also tested the predictability of the four neuromarkers (GM_SAN, FA_CC, fALFF_PFC, and fALFF_pDMN) on other cognitive domain scores shared by both MCCB and CMINDS, including speed of processing, attention,

working memory, verbal learning, visual learning, and reasoning. For each domain score, we built a CMINDS prediction model based on multiple linear regression using the same four neuro-markers generated from the FBIRN cohort as regressors, which is shown in Eq. (2):

$$\text{Cognitive domain scores} = \beta_0 + \text{GM\_SAN} \times \beta_1$$
$$+ \text{FA\_CC} \times \beta_2 + \text{fALFF\_PFC} \times \beta_3 \qquad (2)$$
$$+ \text{fALFF\_pDMN} \times \beta_4$$

The same models are then used to predict the corresponding domain scores in MCCB with UNM and COBRE cohorts. Note that the beta weights $\beta_0$, $\beta_1$, $\beta_2$, $\beta_3$, and $\beta_4$ are different for each domain. Table 1 lists the correlations between the estimated domains scores and true values for all three cohorts. It is clear that all CMINDS domain scores can be predicted by these four neuro-markers. More importantly, such a prediction can be generalized to most of the MCCB domains in two independent UNM and COBRE cohorts, suggesting that the identified neuromarker signatures provide a set of broadly applicable predictors on cognitive performance of either global or specific cognitive domains.

**Contribution of the modality-specific neuromarkers to prediction**. Furthermore, to examine the specific contribution power of each of the four neuromarkers on predicting multiple cognitive domains, we plotted the beta weights ($\beta_1$, $\beta_2$, $\beta_3$, $\beta_4$ as in Eq. (2)) for GM_SAN, FA_CC, fALFF_ PFC, and fALFF_pDMN,

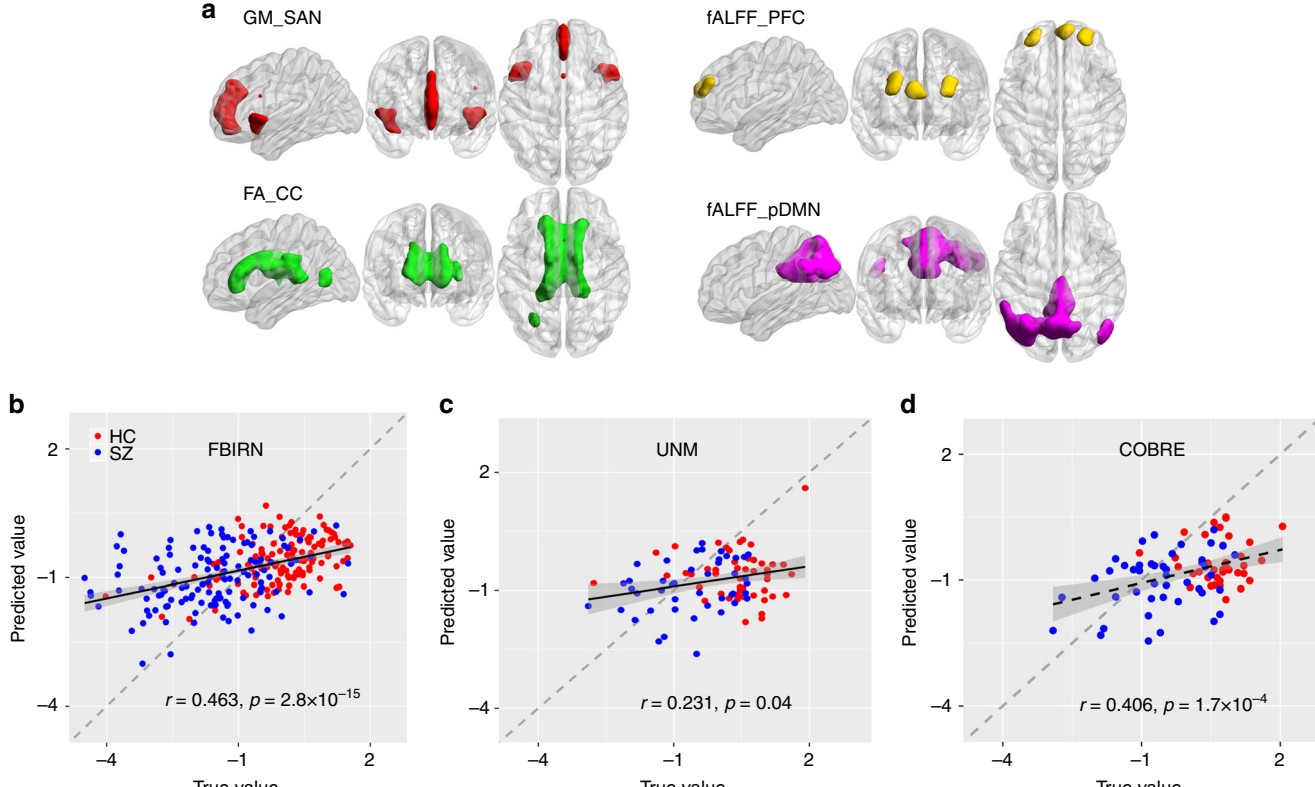

**Fig. 7** Identified multimodal neuromarkers and the predictability on composite cognitive scores across three cohorts. **a** Four identified modality-specific brain networks from FBIRN cohort that were used as regressors to predict individual cognitive scores. **b** Prediction of CMINDS composite scores based on linear regression of the four regressors (mean ROI values in **a**). A correlation of $r = 0.463$ was achieved between the estimated CMINDS composite scores and its true values. **c** Generalization of the CMINDS prediction model in **b** to UNM cohort (41HCs/37SZs) to predict MCCB, $r = 0.231$. **d** Generalization of the CMINDS prediction model in **b** to COBRE cohort (42 HCs/46SZs) to predict MCCB, $r = 0.406$. In both **c** and **d**, good generalizability of the proposed prediction model were validated. The gray regions in **b**–**d** indicate a 95% confidence interval was achieved between the estimated MCCB composite scores and its true values

**Table 1 Prediction results for multiple cognitive domains using the four neuromarkers**

| Predicted measures | CMINDS (FBIRN) | | MCCB (UNM[a]) | | MCCB (COBRE[a]) | |
|---|---|---|---|---|---|---|
| | *r* | *p* | *r* | *p* | *r* | *p* |
| Cognitive composite | 0.463 | 2.8e−15 | 0.231 | 0.04 | 0.406 | 1.7e−04 |
| Speed of processing | 0.470 | 4.8e−16 | 0.206 | 0.05 | 0.351 | 1.3e−03 |
| Attention/vigilance | 0.332 | 3.5e−08 | 0.231 | 0.038 | 0.249 | 0.025 |
| Working memory | 0.402 | 8.0e−12 | 0.218 | 0.05 | 0.230 | 0.039 |
| Verbal learning | 0.371 | 3.8e−10 | 0.230 | 0.04 | 0.370 | 6.7e−04 |
| Visual learning | 0.456 | 5.0e−15 | 0.09 | 0.2 | 0.15 | 0.1 |
| Reasoning/problem solving | 0.330 | 3.5e−08 | 0.193 | 0.08 | 0.190 | 0.07 |

[a]Prediction of MCCB based on the models trained for CMINDS in FBIRN cohort

respectively, as in Fig. 8, which correspond to linear regression results in Table 1.

Evidently, GM_SAN and FA_CC have the higher beta weights for the majority of cognitive domains ($p = 0.002$, ANOVA test on |beta weights| among four neuromarkers). While structural neuromarkers were more predictive and less variable, functional MRI measures were more sensitive to domain differences. This can be reflected by the standard deviations of beta weights across cognitive domains, (i.e., GM_SAN:0.088;FA_CC:0.053; fALFF_PFC:0.130;fALFF_pDMN:0.134), indicating that the contributing power of fALFF shows more fluctuation than GM and FA across different cognitive domains.

## Discussion

In this study, we searched for multimodal neuromarker signatures that can be used to quantify and predict cognitive performance by successive data mining and model generalization. To the best of our knowledge, this is the first attempt to utilize cognition as a reference to guide the three-way multimodal MRI fusion, and to replicate findings in two independent cohorts and across multiple cognitive domains. Our goal was to answer the four challenging issues regarding the cognitive imaging biomarkers. By successfully predicting different measures of cognition in three independent datasets, GM_SAN, FA_CC, fALFF_PFC, and fALFF_pDMN demonstrated great potential as multimodal neuromarker signatures of generalized cognition.

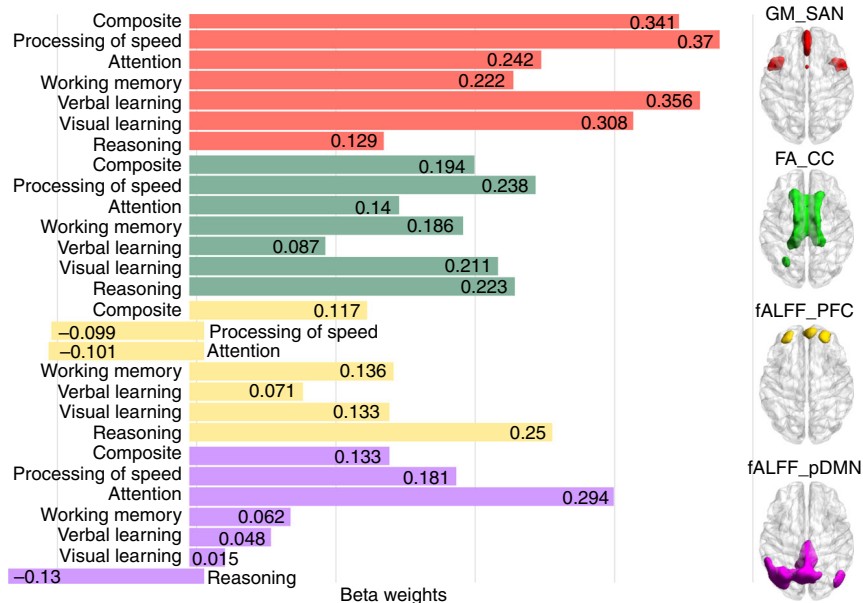

**Fig. 8** Beta weights of GM_SAN, FA_CC, fALFF_PFC, and fALFF_pDMN in predicting multiple cognitive domains. GM_SAN and FA_CC have the higher contribution power for a majority of cognitive domains, while fALFF_pDMN and fALFF_PFC are more sensitive to reflect domain differences (higher standard deviation)

One major finding was that CEN, SAN, and pDMN are three key networks that are especially important for uncovering multiple cognitive functions and their dysfunction in SZ[20–22]. More importantly, they are consistent with the triple network model of major psychopathology[19]. Particularly, in our study, bilateral GM volume reduction has been seen in SAN in schizophrenia patients and were closely associated with all cognitive domain scores. SAN deficits in schizophrenia have been linked to reality distortion, suggest that SAN abnormality may contribute to impaired salience that is associated with hallucinations and delusions in schizophrenia[51]. Together with other consistent findings[51,52], we speculate that SAN in GM may have a crucial role as a structural substrate for neurocognition[19,53]. Moreover, CEN was detected in both fALFF and GM in our results, which is important for maintaining and manipulating information in decision-making[54,55], working memory, and problem solving. In addition, pDMN was identified in fALFF maps related with cognitive composite and working memory domains. These modality-specific triple networks along with FA_CC covary subject-wisely, and were replicated in an independent cohort to be associated with global cognitive function (Fig. 3). Although they have previously been found to be associated with cognitive deficits in multiple reports[38–41], but never in a multimodal joint analysis across multiple cohorts. Whereas our work provides evidence that a synthesis and interaction exist among the three intrinsically coupled networks[19] in both brain function and structure, which are systematically engaged during cognition and impact multiple cognitive domains.

Another interesting finding was that the identified multimodal networks are most significantly correlated with three cognitive domains: attention, working memory, and verbal learning, which belong to key higher cognitive functions and are often severely impaired in SZ[20,21,56]. As seen in our results, working memory is correlated with fALFF and GM, which is in line with CEN (i.e., dlPFC, STG, and PPC) detected in fMRI, and partly in sMRI (dlPFC), where SZ patients showed decreased fALFF and GM values with lower domain scores. This is consistent with the fact that memory shows pronounced deficits in SZ, with working memory primarily affected[22,47]. In addition, attention

encompasses a variety of functions that includes information selection, enhancement of selected information and inhibition of unselected information. We detected GM volume reduction in key nodes of the SAN (bilateral insula and ACC in blue frame as in Fig. 4) and dlPFC associated with attention deficits in SZ. Accordingly, SAN is known to be involved in a variety of monitoring, attention switching, and decision-making processes[50]. Furthermore, regions including AG and hippocampus (in green frame) were commonly detected in fMRI and sMRI, accompanying the domains of verbal learning that correlated with fALFF and GM. Specifically, the Broca's area (BA 44[57]) and posterior Wernicke's area (BA 22[58]) were, respectively, identified in fMRI and sMRI images and both are included in the language learning network. Therefore, the identified multimodal signatures represented a cortical–subcortical circuit in fALFF and GM, which may account for higher cognitive deficits in working memory, attention, verbal learning, and composite cognition in SZ[59]. Furthermore, FA map occurred in ATR (connecting the frontal–subcortical circuits) and SLF (linking frontal–parietal–temporal circuits), implying that alterations in one modality can be linked with changes in distant but connected regions in another modality. This strength in multimodal fusion reveals associations that cannot be discovered by separate multimodal analyses.

Third, our results indicate that structural features are more spatially consistent and replicable across multiple cognitive domains, while functional maps may better differentiate cognitive domains. Specifically, our results revealed one pattern of overlapped GM reduction in SAN extending across four domains (composite cognition, working memory, attention and learning) and in two cohorts. Similar findings were reported, indicating that GM lesions converged in ACC and dorsal-insula (key nodes of SAN)[60] based on the voxel-based morphometry meta-analysis across six diagnostic groups (SZ, bipolar disorder, depression, addiction, obsessive-compulsive disorder, and anxiety), and affected most categories of psychiatric illness[61]. Coincidentally, reduced FA in corpus callosum that links the bilateral hemispheres has been reported in a recently ENIGMA large-scale coordinated study of white matter microstructural differences in

schizophrenia[62], while here we replicated it in a multimodal and multi-domain co-varying manner for the first time. By contrast, fALFF exhibit more spatial variance in cortical regions across a wide range of domains, except with vmPFC and dlPFC being commonly detected that closely associate with high-order cognitive functions, (i.e., working memory[38], verbal learning, and attention[63]); whereas, subcortical regions such as hippocampus, thalamus are more task-specific. We provide new insights into fundamental cognitive imaging biomarkers by identifying co-varying multimodal neuromarker signatures in which functional features are more sensitive to cognitive domain differences, while brain structural and anatomical property show more unifying impaired patterns associated with multi-domain cognitive decline[64].

Finally, the primary goal of neurocognitive imaging studies is to identify neuromarkers that can predict individual educational or health outcomes[8,65]. After discovering a set of stable, domain-shared, multimodal neuromarker signatures (Fig. 7a), we defined a linear regression model (Eq. (1)) that is able to predict individual cognitive composite scores, which can be generalized to two independent cohorts (UNM and COBRE) to predict new unseen subjects on similar but different cognitive measures (Figs. 7 and 8). We have validated the feasibility of the four neuromarkers as predictors of general cognitive performance, suggesting that they are descriptive in nature to cognitive function and may serve as potential biomarkers for cognitive impairments in SZ. Note we obtained high predictive power in this study by selecting four neuromarkers representing common patterns among four domains (Fig. 5). Future work could focus on incorporating domain unique patterns, which we expect to further increase predictive power by better leveraging the sensitivity of the fALFF feature, which showed more domain unique patterns.

A possible limitation of this work is that reference-directed fusion works on extracted features, rather than the original imaging data (e.g., using fALFF instead of 4D fMRI data). Although some of the temporal information was lost using this method, a "feature" tends to be more tractable than working with the large-dimensional original data[66] and provides a simpler space in which to link the data[67]. In future work, we plan to incorporate features such as functional network connectivity matrices[68], dynamic states[69], and structural morphometric measures as fusion input to capture both temporal and spatial co-alterations. Furthermore, most participants were receiving antipsychotic medication at the time of scanning (medication information can be found in Supplementary Table 8 and Supplementary Note 4). In our current study, the correlation between medication dose and cognitive domain was not significant (Supplementary Table 9). In addition, no imaging voxels showed a significant correlation with medication dosage for any of the three modalities, demonstrating that medication has little or at best a very subtle effect on brain imaging in our current data. These results support our claim that the identified replicable multimodal co-varying patterns are associated with cognition but not medication exposure.

In summary, by jointly analyzing three MRI data in a supervised, cognition-guided multimodal fusion model, we successfully identified four multimodal neuromarker signatures that can be replicated in two independent cohorts. A triple brain network including GM_SAN, fALFF_CEN, and fALFF_pDMN may have a crucial role as anatomical substrates of neurocognition, which achieved preferable accuracy for predicting the multi-domain cognitive scores and can be generalized to predict previously unseen individuals. Our results suggest functional features are more sensitive to differentiate cognitive domains, while the salience network in GM and corpus callosum in FA are highly consistent and predictive on multiple cognitive domains. This

work defines modality-specific brain networks that may be broadly applicable as neuromarkers of cognitive impairment, and our assessment can help better understand how functionally and anatomically connected brain systems both engender and constrain cognitive functions in SZ.

## Methods

**Multimodal fusion with reference.** Based on supervised learning, we previously have developed a reference-guided fusion model called MCCAR + jICA[26]. Assume $X_k$ represents multimodal dataset and each is a linear mixture of components $C_k$ with a nonsingular mixing matrix $A_k$, $k = 1, 2, 3$, denoting the modality. Namely, $X_k = A_k C_k$, where $X_k$ is a subjects-by-voxels feature matrix and $A_k$ is a subjects by number of components ($M$) mixing matrix. MCCA with reference (MCCAR) imposes an additional constraint to maximize not only the covariations among loadings of each modality, but also the column-wise correlations between $A_k$ and the reference signal, as shown in Eq. (3).

$$\max \sum_{k,j=1}^{3} \left\{ \left\| \mathrm{corr}\left(A_k, A_j\right) \right\|_2^2 + 2\lambda \cdot \left\| \mathrm{corr}(A_k, \mathrm{ref}) \right\|_2^2 \right\} \quad (3)$$

where ref is an $N \times 1$ vector, denoting the referred measure, $N$ is the subject number. $\mathrm{corr}(A_k, A_j)$ is the column-wise correlation between $A_k$ and $A_j$, and corr $(A_k, \mathrm{ref})$ is the column-wise correlation between $A_k$ and ref. After optimization by MCCAR, we can obtain the potential target components $C_i$ that are correlated with ref in each modality, as well as being most correlated across subjects between modalities. Then joint ICA is further applied to on the concatenated maps of $[C_1, \ldots, C_M]$, in order to keep the modality linkage of the potential target components and maximize the spatial independence. The final independent components (ICs) $S_k$, along with their mixing matrices $D_k$ are obtained by linear source decomposition.

$$W[C_1, C_2, \ldots, C_M] = [S_1, S_2, \ldots, S_M] \quad (4)$$

$$X_k = D_k \cdot S_k = \left(A_k \cdot W^{-1}\right) \cdot S_k \quad k = 1, 2, \ldots, n \quad (5)$$

Among $S_k$, one or more joint independent components (ICs) that are specifically correlated with the ref will be identified as the target $IC_{\mathrm{ref}}$. Therefore, by incorporating prior information, MCCAR + jICA enables identification of a joint multimodal component(s) that has robust correlations within referred measures and amongst themselves (inter-modality correlations), which may not be detected by a blind $N$-way multimodal fusion approach. For more details, please refer to our method paper on MCCAR + jICA[26] and $\lambda$ tuning for Supplementary Note 5.

Here the cognition-guided fusion analyses were implemented by feeding the preprocessed MRI features (i.e., GM density from sMRI, FA from dMRI and fALFF from resting-state fMRI) into MCCAR + jICA, as displayed in Fig. 9a and b. We sought to investigate the target joint independent components ($IC_{\mathrm{ref}}$) that are not only significantly correlated with referred cognitive scores, such as CMINDS composite, MCCB composite or CMINDS domain scores, but also indicated significant linked functional–anatomical–structural alterations between schizophrenia and controls, i.e., group-discriminative.

**Predictive neuromarker extraction.** After identifying multimodal networks of four CMINDS cognitive domains from the FBIRN data (Fig. 9c, d), we extracted the brain regions which were consistently involved within each modality as potential neuromarkers. Take the feature extraction of GM as an example. After converting component GM_$IC_{\mathrm{ref}}$ into Z scores and thresholding at $|Z| > = 2$, masks of GM_$IC_{\mathrm{ref}}$ for each of the four cognitive domains (composite, attention, working memory and verbal learning) were generated. A map of regions included in each of these four GM masks reveals the common ROIs across the four cognitive domains. The final GM mask of GM_SAN is shown in Fig. 7a. This mask of GM was then used to extract ROI features from every subject. The mean of the voxels within the obtained ROI was calculated for each subject, generating a $N_{\mathrm{subj}} \times 1$ feature vector for GM_SAN. The other modalities (dMRI and fMRI) were processed in the same way to get the FA_CC and fALFF_PFC feature vectors. The fourth fALFF_pDMN feature was extracted from fALFF_$IC_{\mathrm{ref\_composite}}$ from the FBIRN data. The resulting regions were those included in the triple network hypothesis[19]. Finally, we formed a feature matrix in dimension of $N_{\mathrm{subj}} \times 4$ for the FBIRN data. For the UNM and COBRE cohorts, the ROI features of each modality were extracted by applying the four masks generated from the FBIRN to the UNM and COBRE data. Following this, the mean of each ROI was calculated for each subject, resulting in a $N_{\mathrm{subj}} \times 4$ feature matrix for the UNM and COBRE cohorts, respectively.

**Multiple linear regression.** After neuromarker extraction, for the FBIRN data, each of the four neuromarker vector was normalized to mean = 0, std = 1. These vectors were then treated as the linear regressors and the corresponding cognitive scores were treated as the targeted measures; together, they were input into the multiple linear regression model to obtain a linear equation for an estimate of the target measures. The same regression model (beta weights) achieved from the

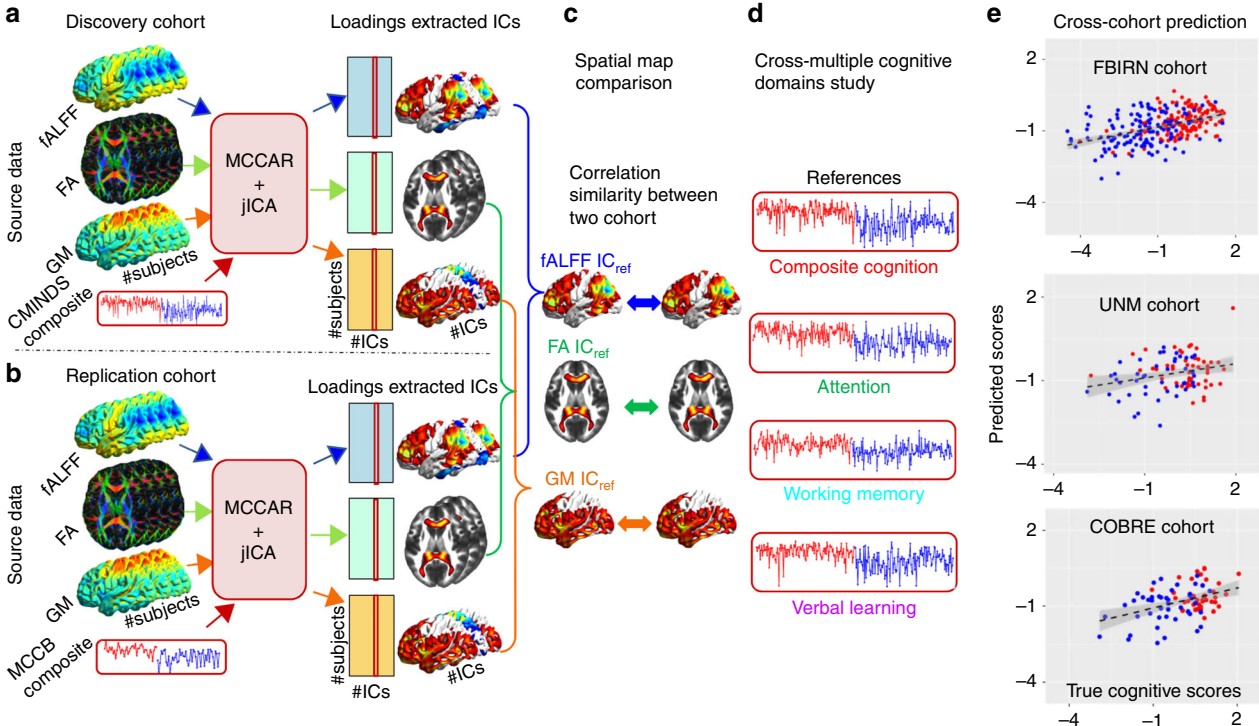

**Fig. 9** Flowchart of cognition-directed multimodal fusion and prediction analysis. First, cognitive scores of **a** CMINDS composite score and **b** MCCB composite score were set as the reference to guide the three-way MRI fusion for the discovery cohort and replication cohort respectively. **c** To test the similarity of the identified neuromarker between cohorts, we compared the permuted spatial correlation between brain maps, and summarized the most affected cognitive domains related to schizophrenic deficit. **d** Furthermore, cognitive domain scores of attention, working memory, and verbal learning were used as reference to guide the three-way MRI fusion respectively, aiming to identify the domain-specific neuromarkers. **e** Finally, after extracting the neuromarker maps across multiple domains, we built multiple linear regression models to predict individualized cognitive scores of FBIRN cohort. The achieved models were further successfully generalized to predict corresponding cognitive measures in two independent cohorts (UNM and COBRE)

FBIRN prediction was then applied to the regressors obtained from UNM and COBRE data. Pearson correlations between true and predicted cognitive scores of each domain were assessed, as seen in Figs. 7, 8 and 9e. Based on the CMINDS regression model obtained from FBIRN cohort, if the estimated MCCB scores of UNM and COBRE cohorts are significantly correlated with the true MCCB scores, the generalization of predictability of cognition based on the four selected neuromarkers can be recognized to some degree. The beta weights of linear regression models for each predicted cognitive domain are displayed in Fig. 8.

**Participants**. Two independent data cohorts were used in this study. One is recruited from FBIRN phase III datasets including 147 SZs (39.5 × 11.7) and 147 HCs (37.4 × 11) that were matched for gender, age, handedness and race distributions. The demographic and cognitive information are summarized in Supplementary Table 10. All subjects were collected from seven FBIRN consortium sites (University of California Irvine, University of California Los Angeles, University of California San Francisco, Duke University, University of North Carolina, University of New Mexico, University of Iowa, and University of Minnesota). Each dataset included diagnosis, age at time of scan, gender, illness duration, symptom scores, and current medications when available, which were shared by each research group according to their site's protocols. Inclusion criteria required all participants to be adults between the ages of 18 and 65 years. Diagnosis of SZ was confirmed by trained raters using the Structured Clinical Interview for DSM-IV (SCID)[70]. All patients were on a stable dose of antipsychotic medication either typical, atypical, or a combination for at least 2 months. Detailed medication information could be found in Supplementary Table 8. Symptom severity was rated using PANSS. All SZs were clinically stable at the time of scanning. In addition, HC participants were excluded for past or current psychiatric illness based on SCID assessment or for having a first-degree relative with a diagnosis of an Axis-I psychotic disorder. Written informed consent was obtained from all participants under protocols approved by the Institutional Review Boards at each study site. Demographic information for subjects of each site is provided in Supplementary Table 11.

The UNM cohort consisted of 39 SZs (35.6 ± 13.1) and 44 HCs (36.3 ± 12.5) who were collected from the University of New Mexico and recorded with MCCB scores. The third cohort include 42 patients with schizophrenia (39.3 ± 13.2) and 42 HCs (40 ± 11) who were collected from the COBRE project[28]. Details of the

cognitive and clinical information of both UNM and COBRE data can be found in Supplementary Table 12 and Supplementary Table 13.

**Imaging parameters**. The resting-state fMRI data for the FBIRN cohort was collected on six 3T Siemens and one 3T General Electric (GE) scanner. The imaging protocol for the resting-state scans at all sites was the same: a T2∗-weighted AC-PC aligned echo planar imaging sequence (TR = 2 s, TE = 30 ms, flip angle = 77°, 3.4 × 3.4 × 4 mm with 1 mm gap, 162 frames, 5:38 min). For the resting scan, subjects were instructed to lie still with eyes closed.

DMRI data for the FBIRN cohort were also acquired on six 3T Siemens and one 3T GE. All parameters for these two scanners were the same except for TE (Siemens: 84 ms, GE: 81.7 ms). The rest of the parameters for both Siemens and GE were as follows: TR = 9 s; field of view (FOV) = 256 × 256 mm; slice thickness = 2 mm; number of slices = 72; slice gap = 2 mm; voxel resolution 2 × 2 × 2 mm; flip angle = 90°; number of diffusion gradient directions = 30, b = 800 seconds/mm², and 5 measurements with b = 0. All images were registered to the first b = 0 image by FMRIB Linear Image Registration Tool (FLIRT: http://fsl.fmrib.ox.ac.uk/fsl/fslwiki/FLIRT).

High-resolution T1-weighted brain imaging data for the FBIRN cohort were also acquired on six 3 T Siemens and one 3T GE. Siemens scan parameters were TR = 2.3 s, TE = 2.94 ms, flip angle = 9°, resolution = 256 × 256 × 160. GE scan parameters were TR = 5.95 s, TE = 1.99 ms, flip angle = 12°, resolution = 256 × 256 × 166. All scans covered the entire brain with FOV = 220 mm², voxel size = 0.86 × 186 × 1.2 mm³.

**Data preprocessing**. The fMRI data were preprocessed using the automated analysis pipeline in SPM8 (http://www.fil.ion.ucl.ac.uk/spm) as following: motion correction, slice timing and normalization to MNI space, including reslicing to 3 × 3 × 3 mm voxels. We removed subjects who had framewise displacements (FD) exceeding 1.0 mm, and head motion exceeding 2.0 mm of maximal translation (in any direction of x, y, or z) or 1.0° of maximal rotation throughout the course of scanning. We also despiked the fMRI data, regressed out six head motion parameters, white matter, and cerebrospinal fluid in the denoising procedure. Results indicate FD (mean framewise displacements, mean of root of mean square frame-to-frame head motions assuming 50 mm head radius) for all subjects were <0.3 mm at every time point. There is no significant difference between patients and controls on mean FDs; namely, UNM, HC: mean = 0.22 ± 0.12 mm, SZ: 0.21 ± 0.11 mm,

two sample t-test: $p = 0.77$, FBIRN, HC: mean $= 0.25 \pm 0.18$ mm, SZ: $0.27 \pm 0.21$ mm, two sample t-test: $p = 0.65$. No imaging voxels showed a significant correlation with mean FD after the FDR multiple comparison correction for any of the three modalities, and also no significant correlations between mean FD and cognition. Detailed head motion correction could be found in the Supplementary Note 6. Data were then spatially smoothed with an 8 mm, full-width half-maximum (FWHM) Gaussian filter. The sum of the amplitude values in 0.01 to 0.08 Hz low-frequency range was divided by the sum of the amplitudes over the entire power spectrum[48] to obtain fALFF. Finally, the fusion analysis was conducted on the spatial maps of fALFF. Considering there is no group difference in head motion, no significant correlations between mean FD and cognitive scores, and the correlations with cognitive scores are still significant after regressing out mean FD, we believe that micro-motion is not a major factor affecting the current results.

The dMRI data were preprocessed using the FMRIB Software Library (www.fmrib.ox.ac.uk/fsl) and consisted of the following steps. Firstly, quality check and remove gradient directions with excessive motion or vibration artifact. Secondly, motion and eddy correction. Thirdly, correction of gradient directions for image rotation due to the motion correction procedure. Finally, FA was calculated and smoothed using an 8 mm FWHM Gaussian filter.

The sMRI data were spatially normalized to MNI space using the unified segmentation method in SPM8, segmented into GM, white matter, and cerebral spinal fluid (CSF). Then, the GM volume were smoothed with an FWHM of 8 mm Gaussian filter. Subject outliers were further detected by spatial Pearson correlation with the template image to ensure that all subjects were segmented properly.

**Normalization and site effect correction**. After preprocessing, the three-dimensional brain images of each subject were reshaped into a one-dimensional vector and stacked, forming a matrix ($N_{subj} \times N_{voxel}$) for each of the three modalities. These three matrices were then normalized to have the same average sum of squares (computed across all subjects and all voxels for each modality) to ensure all modalities had the same ranges. To ensure that each of the modality features were not confounded by site-related differences in subject recruitment criteria or by other unidentified variables, multivariate analysis of covariance (MANCOVA) was performed on all feature matrices. Site, gender, age, and their interactions were all regressed out from GM, fALFF, and FA features, respectively, to minimize their impact on the brain imaging data. Thus, the resulting data were then ready for fusion analysis.

**Data availability**. The code for the supervised fusion algorithm has been released and integrated in the Fusion ICA Toolbox (FIT, https://mialab.mrn.org/software/fit), which can be downloaded freely and used directly by users worldwide. The multimodal data used in the present study can be accessed upon request to the corresponding authors.

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

## Acknowledgements

This work was supported by the Chinese National Natural Science Foundation Nos. 81471367, 61773380, and 61703253, the Strategic Priority Research Program of the Chinese Academy of Sciences (Grant No. XDB02060005), the National High-Tech Development Plan (863 program, No. 2015AA020513), "100 Talents Plan" of Chinese Academy of Sciences, and the NIH grants R01EB005846, 1R01MH094524, and P20GM103472 as well as NSF grant 1539067.

## Author contributions

J.S. and V.D.C. designed the study; S.Q. performed the data analysis; J.S., S.Q., and V.D. C. wrote the paper. J.B., T.G.M.v.E., J.A.T., D.H.M., J.M.F., J.V., B.A.M., A.B., S.M., S.G. P., and A.P. contributed to the multimodal imaging and cognitive data. E.D., Z.F., and R. J. helped with data preprocessing. All authors contributed to the results interpretation and discussion.

## Additional information

**Competing interests:** The authors declare no competing interests.

