## [Peer Review File · Nature Communications]

Reviewers' comments:

Reviewer #1 (Remarks to the Author):

Calhoun and team report a 3-way multimodal ICA-based data fusion in a large cohort of patients with schizophrenia and healthy controls. They also make an attempt to replicate their observations in an independent cohort (n=83), which is an effort that is highly commendable. Their observations implicate the salience network grey matter, corpus callosum's fractional anisotropy, and central executive and default mode networks' amplitude of low-frequency resting fluctuations as the markers of generalized cognitive ability across subjects.

This is a very well designed analysis by a highly respected analytical team. It will come across as complex to many readers as the methods used are not still in widespread use in the imaging community. Given the complexity of analysis, the dependence of findings on derived rather than absolute MRI measures, and lack of a convincing replication of the referenced ICs in the independent cohort (see below for further clarification), the translational utility as claimed by the authors may not materialize. Nevertheless, this is an observation that adds a nuanced perspective to the existing literature of the neural-network basis of cognitive defects in schizophrenia.

I was struck by how the authors chose to completely ignore the issue of antipsychotic dose exposure – both cumulative (lifetime) and cross-sectional dosages are linked to GM loss, fMRI changes as well as FA changes. To make matters worse, a large proportion of variance in measured cognition in schizophrenia may be affected by the dose of antipsychotics taken by an individual. The 'reference' factor binding the components reported here may very well be medication exposure, not cognition. Please refer to the works of Nancy Andreasen, and the meta-analysis by Knowles and David.

Also, the authors parsed individual domains of cognition only in relation to the ICs that related to general cognitive composite. This unnecessarily constrains the analysis in such a way that the brain regions that independently relate to a domain e.g. visual memory but not global composite, will not be identified in the search. This may improve replicability at a serious cost of validity. Please refer to the works of Dwight Dickinson in this regard (esp reg global and specific domains of cognition).

The term 'fMRI components' is better replaced by 'fALFF components' wherever possible so a causal reader does not equate this with connectivity metrics.

Given the major findings pertain to SN, I do not find a well referenced discussion regarding the role played by the SN in schizophrenia. The insular dysfunction hypothesis was developed around 2010, with several structural findings relating SN deficits to symptoms such as reality distortion.

I am not fully convinced that the replication across the data sets is excellent. The peak coordinates for every modality replicate poorly across the samples. The claim seems to be based on spatial correlation. Are the spatial correlation values based on binary conjunction maps thresholded at a specific Z value or simple voxel-by-voxel correlation of component loading across the 2 datasets? If it is the former, then the size of reported r will depend greatly on the Z cut-off.

I am generally enthusiastic about publishing this work. But I would like to see a more realistic estimate of translational utility that takes into consideration the complexity of the analysis and issues with replication. I would also like to see more focus on mechanistic clarifications offered by the reported observation.

Reviewer #2 (Remarks to the Author):

In "Searching for multimodal neuromarkers in schizophrenia via cognition guided MRI data fusion: a cross-cohort and multi-domain study," Sui et al. describe multimodal neuromarkers of cognition in a sample of healthy control samples and patients with schizophrenia. While this study has a good deal to recommend it, including a very interesting supervised data fusion method, there are several aspects of this work which at present diminish its impact.

1. Style. Despite great interest, I found the paper fairly difficult to follow, which was largely due to a combination of a fairly convoluted analytic (see below) flow and a somewhat cumbersome writing style. For example, the abstract is poorly written, laden with run-on sentences and incomplete information-- "cognitive composite and multiple domain scores were used to guide three-way multimodal MRI data fusion in a discovery cohort(n=294) respectively."

2. Sample and inference. The authors seek to develop a multi-modal dimensional biomarker of cognition. However, this is accomplished in a mixed group of healthy controls and patients with schizophrenia. Because of the large cognitive deficits in patients, the neuromarkers discovered tend to be an admixture of the effects of diagnosis and individual differences in cognition. At one point the authors mention controlling for diagnosis (via partial correlation)—this should be done through the paper, which would allow the authors to show that dimensional effects are present in both groups.

3. Methods description. The readability of the manuscript would be improved if some of the relevant methods content could be placed within the context of the results section, especially the Multimodal fusion with reference subsection, to provide intuitions of the MCCAR+jICA throughout the manuscript.

4. Parameter tuning. It is difficult to tell, but I imagine regularization is occurring to handle the $p \gg n$ problem of the data used. How regularization is performed, what constraints are used, and how these parameters are tuned are not discussed in the methods. (For example, there is a λ in one of the equations that is not defined). If parameter tuning is performed, how do the authors ensure that they are not over-fitting the data? If significant tuning is occurring, this method should be cross-validated.

5. Significance testing. Relatedly, it is unclear what the null distribution of this method is. If significant regularization is occurring, it is quite possible that the data is over-fit and the null distribution is not centered at zero. Standard permutation testing techniques that show the null, and how the observed correlations compare to the null, would increase confidence in the results substantially.

6. Use of replication sample. It was somewhat curious how the authors used the replication sample. Rather than simply applying the model found from the discovery sample, new features were found de-novo. How well does the model from the discovery sample predict general cognition in the replication sample?

7. Statistical testing of features in replication sample. One of the ways in which the authors assess the replication sample is through spatial correlations of the loading maps. They report high correlations, which are reasonably persuasive. However, statistical tests are also reported, which are uniformly so low as to be below the precision of the test ($p < 10^{-20}$, etc). These values are meaningless, as they seem to consider voxels as "samples" in the correlation. Thus, by simply increasing the resolution of the images in the spatial correlation, the authors are gaining additional "statistical power", which is arbitrary and based purely on re-sampling. To test these relationships appropriately, a spatial permutation procedure that preserves the structure of the data is necessary—this has frequently been done on a spherical surface (see Gordon et al., Cerebral

Cortex 2014 for example).

8. Feature overlap. The rationale for finding overlapping features across cognitive sub-domains and then testing them separately was very unclear. Why not simply use the original features defined by the data fusion technique?

9. Resting state pre-processing. The authors used a de-noising pipeline which certainly leaves a huge amount of noise related to motion and physiological artifact in the data, but do not control for motion in the group level analyses using fALFF based components. This of course can be a substantial confound in group level analyses—see for example recent data regarding this risk from the HCP below. Please display the group level correlation between fALFF neuromarkers and mean FD, cognition and FD, and control for FD in group level analyses.

Siegel, J.S., Mitra, A., Laumann, T.O., Seitzman, B.A., Raichle, M., Corbetta, M., and Snyder, A.Z. (2016). Data Quality Influences Observed Links Between Functional Connectivity and Behavior. *Cereb Cortex*

Minor

1. Typo/Grammar Page 17. "Except for" should be "in addition to", or "besides"
2. Clarify in figures Figure 7 d-e. No labeling of the panels were provided to indicated that these data were from FBIRN alone, not from UNM or combined dataset.

Reviewer #3 (Remarks to the Author):

In this manuscript, Sui and colleagues tested whether multimodal MRI data can be leveraged to provide useful predictors of cognitive ability in healthy controls and cognitive impairment in schizophrenia. This topic is of keen interest to the brain imaging community, and this study is noteworthy for its use of multimodal data and cross-validation in an independent dataset. The impact of this manuscript hinges on the generalizability of the identified multimodal biomarkers. On this, my enthusiasm is tempered by the following concerns.

Main points:

The authors have not given sufficient consideration for head motion as a factor that affects image quality in all three MR modalities included. The influence of head motion needs to be addressed, e.g. by accounting for differences in motion between individuals and particularly between patient and control groups. First, more aggressive removal of motion signals could have been applied to the data (e.g. by regression of whole-brain signal). Second, summary motion parameters should be included in the multiple regression as a covariate. Third, a useful proof that motion does not determine the results might come from using the motion parameters as the reference in the "MCCAR+jICA" clustering and repeating the analyses. Individual differences in head motion could be a uniting factor that links the observed imaging features in the different modalities.

It would be interesting to see a null model – for instance using a permuted vector as reference – to give us an indication of what kinds of spatial patterns we might expect by chance. This would give us confidence that the observed patterns are specific to the cognitive measures being used as priors. I found it interesting that the areas identified in the fMRI and FA data were regions that typically show the highest signal in each of those modalities. i.e. the corpus callosum and corticospinal tracts typically show high FA, and the default mode network typically exhibits high signal in fMRI. Is it possible that these regions were clustered together due to their SNR properties, rather than any interesting associations with cognitive measures?

I found the generalization of the derived model to be underwhelming. Evidence for generalization

came from two forms, 1) the spatial similarity of the IC maps defined separately in the two datasets, and 2) application of the regression model derived from one dataset to the other:

1) Spatial similarity: The spatial maps derived from the two datasets were correlated to ~ 0.5 , however the details of how the spatial correlation was performed are not fully explained. Were the spatial maps masked to exclude white matter voxels? For instance, in the GM correlation in Fig 3, were all voxels included in the final correlation analysis, or was the correlation restricted to the GM voxels? If the former, this would inflate the correlations. More information about the spatial correlation would be welcome.

2) Regression model: The high p values in Figs 1 and 7 and Table 1 suggest that there is a large degree of overfitting happening in the training dataset. Hence the frequent reporting of p-values from the training dataset is misleading as the real test of the model's validity is its generalizability to the replication dataset (i.e. Figure 7c). Here, the model fit is not very good ($r=0.2$, $p=0.04$), and to my eye there is no clear relationship between True and Predicted values. It looks to me like the slope is influenced by the red datapoint at the top right of the plot, and I would need to see the same plot with outliers removed before I could be convinced of any relationship.

On a similar note, if the composite cognition scores were used as a reference for the definition of the components, how informative is it to report a p value for the correlation between component loadings and composite cognition scores in Figure 1? Is it surprising that there is a good fit of the loadings to the scores?

As a general point, I found the motivation of the study slightly muddled. The stated aim is to unearth markers of cognitive ability, however the authors also study cognitive impairment in schizophrenia. Why are these two aims carried out in the same datasets? Does the inclusion of schizophrenic patients determine the results? And if so, doesn't this make the identified markers less generalizable indicators of normal cognitive ability? Although the authors did test that the model holds when patients were excluded, the model was still derived using both patients and controls.

Other points:

Abstract:

The term "neuromarker" is defined in the introduction, after being mentioned in the abstract.

The inclusion of schizophrenic patients in the dataset is not mentioned in the abstract.

Introduction:

P3 > Definition of "neuromarker": The term "neurocognitive" seems misplaced here.

P3 > Grammar: 'For example, Rosenberg et al., demonstrate that a set of whole-brain functional network strength'

P3 > "Cognitive Composite ability" – Please explain how this composite measure is defined.

Aim 1 – Given the multimodal nature of the analysis, how do the authors know a priori that the "identified multimodal neuromarker" will be a 'network', classically defined based on anatomical connectivity, or spontaneous correlations between brain regions, or perhaps even distributed regions showing similar task activation profiles? The authors need to define what they mean by terms like 'network' and 'neuromarker' more carefully. For instance, in the Results section the term "multimodal neuromarker network" is introduced. Are these 'networks' only definable by a confluence of imaging techniques? How does this then relate to the "brain networks" as defined classically?

Aim 3 – Similarly, “What are the neuromarker networks associated with specific representative cognitive domains?” > By the authors definition aren’t the “neuromarker networks” defined by their association with cognitive domains? Perhaps remove the word “neuromarker”?

Aim 3 – What is a “cognitive domain discrepancy”?

Given that the central hypothesis of the study follows from the Triple Network Theory proposed by Vinod Menon, it would be useful to have a summary of this in the introduction, particularly in regards to the roles of the three networks in question, and schizophrenia.

P4 > “...each of the four cognitive domains...” > the domains being referred to have not yet been defined.

P4 > Grammar: “Remarkably, these models were successful in predicting for new individuals in an independent cohort on the corresponding cognitive metrics. This validates the generalizability of the identified modality” > “Remarkably, these models were successful in predicting the corresponding cognitive metrics for new individuals in an independent cohort”.

Methods:

P25 > Typo: “we can obtained”

P25 > Please state explicitly what A_j denotes in equation 3.

P25 > Please provide more details about how the target ICref is defined. Is the component that has highest correlation with the ref selected?

P26 > Typo: “After defined”

P26 > The section titled: “Predictive neuromarker extraction” was hard to follow. At one point we are referred to Figure 7a for explanation of the methods.

Figures:

Fig 1: Typo: “congnition”

Dear editor:

Thanks for giving our article further consideration. We have performed a number of additional analyses and revised the paper thoroughly based on the constructive comments made by the reviewers. Below, we shown the reviewers' comments in black and the responses to the comments in blue. For convenience, we have also included the updated text added to the manuscript which is underlined.

Comments to the Authors:

Reviewer #1 (Remarks to the Author):

Calhoun and team report a 3-way multimodal ICA-based data fusion in a large cohort of patients with schizophrenia and healthy controls. They also make an attempt to replicate their observations in an independent cohort (n=83), which is an effort that is highly commendable. Their observations implicate the salience network grey matter, corpus callosum's fractional anisotropy, and central executive and default mode networks' amplitude of low-frequency resting fluctuations as the markers of generalized cognitive ability across subjects.

This is a very well designed analysis by a highly respected analytical team. It will come across as complex to many readers as the methods used are not still in widespread use in the imaging community. Given the complexity of analysis, the dependence of findings on derived rather than absolute MRI measures, and lack of a convincing replication of the referenced ICs in the independent cohort (see below for further clarification), the translational utility as claimed by the authors may not materialize. Nevertheless, this is an observation that adds a nuanced perspective to the existing literature of the neural-network basis of cognitive defects in schizophrenia.

Thank you for the useful and insightful comments.

1. I was struck by how the authors chose to completely ignore the issue of antipsychotic dose exposure – both cumulative (lifetime) and cross-sectional dosages are linked to GM loss, fMRI changes as well as FA changes. To make matters worse, a large proportion of variance in measured cognition in schizophrenia may be affected by the dose of antipsychotics taken by an individual. The 'reference' factor binding the components reported here may very well be medication exposure, not cognition. Please refer to the works of Nancy Andreasen, and the meta-analysis by Knowles and David.

Thanks for highlighting this important point. Not surprisingly, most of the patients enrolled in our current study were taking antipsychotic medications. In response to the reviewer, we performed correlation analysis between cognitive domain scores and medication dosages. A standardized total dose of drug dose, i.e., Chlorpromazine equivalent doses¹, were used to estimate medication dose. Supplementary Table 9 list the *p* values for correlations with all cognitive domains. It is clear that there is very little association between medication dose and cognitive scores in our current data.

Table 9. Correlation analysis between medication dosages and cognitive scores

Cognitive domains	Composite	Speed of Processing	Attention	Working Memory	Verbal Learning	Visual Learning	Reasoning
p value	0.571	0.439	0.768	0.096	0.398	0.772	0.442
r	0.054	0.075	0.029	0.121	0.083	-0.028	0.074

We also performed correlation analysis between medication dosages and multimodal imaging features (voxel-wise). No imaging voxels showed a significant correlation with medication dose, again suggesting that medication dose have little or at best very subtle effects on the brain imaging. These results support our claim that the identified replicable multimodal covarying patterns are associated with cognition but not medication exposure.

We have added the following contents in discussion on page 26.

“...Furthermore, most participants were receiving antipsychotic medication at the time of scanning (medication information can be found in Supplementary Table 8). In our current study, the correlation between medication dose and cognitive domain was not significant (Supplementary Table 9). In addition, no imaging voxels showed a significant correlation with medication dosage for any of the three modalities, demonstrating that medication has little or at best a very subtle effect on brain imaging in our current data. These results support our claim that the identified replicable multimodal covarying patterns are associated with cognition but not medication exposure.”

2. Also, the authors parsed individual domains of cognition only in relation to the ICs that related to general cognitive composite. This unnecessarily constrains the analysis in such a way that the brain regions that independently relate to a domain e.g. visual memory but not global composite, will not be identified in the search. This may improve replicability at a serious cost of validity. Please refer to the works of Dwight Dickinson in this regard (esp reg global and specific domains of cognition).

The primary purpose of the study is to investigate the multimodal imaging patterns associated with global cognitive deficits. Beyond this, we further identified the imaging patterns that are specifically associated with 3 subdomain scores including attention, working memory and verbal learning (as shown in Fig. 5-6). Therefore, investigation of other cognitive domain can be easily generalized by our current framework, while we choose these 3 domains mainly due to that they were most impaired in SZ and closely relevant to global composite.

In addition, the global cognitive composite is also significantly correlated with all other sub-domain scores. A variety of statistical analyses have showed that individual test scores and domain specific scores were significantly inter-correlated and that domain specific scores were highly correlated with the global composite score², as listed in the last column of Supplementary Table 10 (FBIRN-CMINDS) and Table 11 (UNM-MCCB). The work of Dwight Dickinson (Dickinson D, Harvey PD.2009.“Systematic hypotheses for generalized cognitive deficits in schizophrenia: a new

take on an old problem”. *Schizophrenia bulletin* 35, 403-414), also argue that a generalized cognitive deficit is at the core of schizophrenia and deserves more focused consideration from cognitive specialists in the field².

Table 10. Demographics and the CMINDS scores for FBIRN subjects

Measure	HC	SZ	p	r	
Number	147	147			
Age	37.4±11.1	39.5±11.8	0.117	-0.303	
Gender	44F/103M	35F/112M	0.238	-0.139	
CMINDS	Composite	-0.017±1.0	-1.590±1.2	1.7E-24	1
	Speed of processing	-0.010±1.0	-1.356±1.1	2.3E-21	0.729
	Attention/vigilance	0.002±1.0	-1.435±1.4	2.7E-18	0.770
	Working memory	0.010±1.0	-1.152±1.1	1.9E-17	0.731
	Verbal learning	0.024±1.0	-1.373±1.2	1.1E-21	0.785
	Visual learning	-0.017±1.0	-1.051±1.1	1.5E-13	0.830
	Reasoning/problem solving	-0.034±1.0	-0.803±1.2	6.6E-08	0.663
PANSS	Negative	NA	14.556±5.6	-0.256	
	Positive	NA	15.424±4.8	-0.121	

* *r* means correlation value with the CMINDS composite score.

Table 11. MCCB domain scores and PANSS scores of the UNM subjects

Measure	HC	SZ	p	r	
MCCB	Composite	50.4±10.6	30.5±16.1	2.5E-08	1
	Speed of processing	52.1±9.2	34.5±14.4	2.2E-08	0.912
	Attention/vigilance	49.0±10.3	35.7±15.1	2.8E-05	0.864
	Working memory	46.9±11.4	35.8±14.8	3.8E-04	0.839
	Verbal learning	47.9±9.3	38.2±9.1	1.0E-05	0.810
	Visual learning	49.2±9.1	36.8±12.7	5.6E-06	0.787
	Reasoning/problem solving	48.8±9.3	36.8±12.7	10.0E-06	0.787
	Social cognition	54.8±9.8	45.8±11.4	0.5E-04	0.614
PANSS	Negative	NA	28.923±11.4	-0.392	
	Positive	NA	15.846±5.5	-0.488	

* *r* means correlation value with the MCCB composite score.

3. The term ‘fMRI components’ is better replaced by ‘fALFF components’ wherever possible so a causal reader does not equate this with connectivity metrics.

Thank you for the suggestion. We have changed the term “fMRI components” to “fALFF components” throughout the manuscript.

4. Given the major findings pertain to SAN, I do not find a well referenced discussion regarding the role played by the SAN in schizophrenia. The insular dysfunction hypothesis was developed around 2010, with several structural findings relating SAN deficits to symptoms such as reality distortion.

Thank you for the suggestion. We have added the following contents to describe the role of SAN

in schizophrenia. See also manuscript page No. 22-23.

“Both functional and structural studies have pointed to a dysfunctional SAN in schizophrenia³. Bilateral volume reduction has been seen in the anterior insula and ACC in patients with schizophrenia. Furthermore, SAN deficits in patients with schizophrenia have also been linked to reality distortion, leading to the suggestion that SAN abnormality leads to an impaired attribution of salience to stimuli that is associated with delusions and hallucinations in schizophrenia^{3,4,5}. Together with other consistent findings⁶, we speculate that SAN in GM may play a crucial role as a structural substrate for neurocognition^{7,8}. In particular, the insula (an integral hub for initiating network switching that leads to the functional engagement of the CEN and functional disengagement of the pDMN) and its deficit may impact across multiple cognitive domains. Within this hierarchy, the SAN would stand in a ‘hub’ position at a ‘crossroads’ within the functional architecture of the brain, acting as a switch to deploy other major functional networks according to motivational demands and environmental constraints⁹.”

- [3] Dickinson D, Harvey PD. Systemic hypotheses for generalized cognitive deficits in schizophrenia: a new take on an old problem. *Schizophrenia bulletin* **35**, 403-414 (2009).
- [4] White TP, Joseph V, Francis ST, Liddle PF. Aberrant salience network (bilateral insula and anterior cingulate cortex) connectivity during information processing in schizophrenia. *Schizophrenia research* **123**, 105-115 (2010).
- [5] Palaniyappan L, Mallikarjun P, Joseph V, White TP, Liddle PF. Reality distortion is related to the structure of the salience network in schizophrenia. *Psychological medicine* **41**, 1701-1708 (2011).
- [6] Sridharan D, Levitin DJ, Menon V. A critical role for the right fronto-insular cortex in switching between central-executive and default-mode networks. *Proceedings of the National Academy of Sciences of the United States of America* **105**, 12569-12574 (2008).
- [7] Menon V, Uddin LQ. Saliency, switching, attention and control: a network model of insula function. *Brain structure & function* **214**, 655-667 (2010).
- [8] S. Kristian Hill, James L. Reilly, Richard S.E. Keefe, James M. Gold, Jeffrey R. Bishop, Elliot S. Gershon CAT, Godfrey D. Pearlson, Matcheri S. Keshavan, John A. Sweeney. Neuropsychological Impairments in Schizophrenia and Psychotic Bipolar Disorder: Findings from the Bipolar-Schizophrenia Network on Intermediate Phenotypes (B-SNIP) Study. *Am J Psychiatry* **170**, 1275–1284 (2013).
- [9] Menon V. Large-scale brain networks and psychopathology: a unifying triple network model. *Trends in cognitive sciences* **15**, 483-506 (2011).

5. I am not fully convinced that the replication across the data sets is excellent. The peak coordinates for every modality replicate poorly across the samples. The claim seems to be based on spatial correlation. Are the spatial correlation values based on binary conjunction maps thresholded at a specific Z value or simple voxel-by-voxel correlation of component loading across the 2 datasets? If it is the former, then the size of reported r will depend greatly on the Z cut-off.

In response to the reviewer, we have added the following figure to compare the peak coordinates of components between FBIRN and UNM for all three modalities in addition to Fig. 3. Notably, the anatomic locations of the peak coordinates replicate between the 2 datasets and includes caudate and

the bilateral insula for GM, precuneus for fALFF, and posterior and anterior corpus callosum for FA.

Peak coordinates comparison between FBIRN and UNM for three modalities.

The original spatial correlation is based on the voxel-by-voxel correlation of the thresholded component, but not the binary conjunction maps. As suggested by reviewers, we now calculated the spatial correlation of the identified target component between two cohorts using only voxels masked at $|Z| > T$ (threshold). First, the spatial maps were transformed into Z scores and masked at $|Z| > T$. Then we obtained two masks from FBIRN (mask_FBIRN) and UNM (mask_UNM) respectively, which were used to perform the voxel selection. Only voxels that fell in the union of the masks (mask_FBIRN \cup mask_UNM) were used to calculate the cross-cohort correlation. Thus total number of voxels for calculating the spatial correlation is greatly reduced, *e.g.*, from $n=153594$ (whole brain voxels) to $m=1936$ ($T=2$, used in our paper). Spatial correlation was finally performed on these commonly identified voxels ($m=1936$) between two cohorts.

As suggested, we further compared the impact of using different T thresholds on cross-cohort spatial correlations. As listed in Table 4, all cross-cohort correlations r are significant (FDR corrected) regardless of different T thresholds, with $p < 1.0e-5$ in all cases. We have added the above content and the Table 4 to supplementary file in section of “cross-cohort spatial correlation”.

Table 4 Spatial correlation derived from different thresholded T values

Threshold	GM		FA		fALFF	
$T = 1$	$r = 0.38^*$	$m = 8553$	$r = 0.42^*$	$m = 7738$	$r = 0.22^*$	$m = 16420$
$T = 2$	$r = 0.51^*$	$m = 1936$	$r = 0.59^*$	$m = 2720$	$r = 0.39^*$	$m = 3692$
$T = 3$	$r = 0.65^*$	$m = 405$	$r = 0.67^*$	$m = 845$	$r = 0.45^*$	$m = 732$

In addition, we also performed a permutation test to calculate the significance for the cross-cohort spatial correlation. We do this by randomly shuffling Y (UNM_IC_{ref}) across voxels and re-running the correlation analyses (between X [FBIRN_IC_{ref}] and Y) 10000 times in order to obtain an empirical null distribution. We then record the number of times the correlation exceeds the obtained sample correlation. Take GM component for example (Fig. 3), the observed correlation between FBIRN_IC_{ref} and UNM_IC_{ref} was 0.51, while 8 of the 10000 permutations obtained correlations falling out the range of $[-0.51, 0.51]$, thus the probability of $p=8.0 \times 10^{-4}$ was estimated for cross-cohort correlation of $r=0.51$ between GM maps by chance.

We have updated Fig. 3 as below by replacing the original p values with permuted p values in the

blue arrows (cross-cohort replication). The corresponding contents was also added in Supplementary section “Cross-cohort spatial correlation”. This could respond to **Reviewer 2 point #7** as well.

Figure 3. Similarity and covarying patterns of the identified neuromarker networks between cohorts. (a) The identified neuromarker network associated with CMINDS composite in FBIRN cohort. (b) The identified neuromarker network associated with MCCB composite in UNM cohort. The spatial maps were visualized at $|Z| > 2$; The blue arrows represent the cross-cohort spatial correlation (P_{perm} are permuted p values of 10000 randomizations); the orange arrows represent inter-modality covariation across subjects; and the pink arrows denote the correlations between cognitive composite (the reference) and the identified $IC_{ref_composite}$.

6. I am generally enthusiastic about publishing this work. But I would like to see a more realistic estimate of translational utility that takes into consideration the complexity of the analysis and issues with replication. I would also like to see more focus on mechanistic clarifications offered by the reported observation.

As to a realistic translational perspective, we identified a replicable multimodal covarying pattern, including GM_SAN, FA_CC, fALFF_PFC and fALFF_pDMN, which define a modality-specific brain network that may be used broadly as neuromarkers, i.e., to predict multiple cognitive domain scores for unseen subjects in independent datasets, even in the case where the cognitive scores were measured from two different methodologies (CMIND vs. MCCB, they are two similar but not identical cognitive measurement systems¹⁰). We have modified the Discussion as follows on page 23-24:

“Neuroimaging techniques like structural and functional magnetic resonance imaging have led to a search for neuromarkers that can bring psychiatry from subjective descriptive classification into objective and tangible brain-based measures^{11, 12}. The four neuromarkers (including GM_SAN,

FA_CC, fALFF_PFC and fALFF_pDMN) identified in our current work can be applied directly to predict cognitive ability in new individuals and similarly potentially for other mental disorders (such as ADHD or MDD). In addition to cognitive dysfunction, other clinical measures could also be studied using the analysis pipeline proposed in our current study including symptom severity, intelligence quotient (IQ), and behavioral measures (e.g., temperament inventory, reading ability), or even epigenetic variants (e.g., the expression levels of a specific microRNA¹³), suggesting a wide utility in the neuroimaging community. This approach is consistent with the scope of the research domain criteria (RDoC) project proposed by NIMH as well, which aims to incorporating genetics, neuroimaging and cognitive science into classifying and clarifying the underlying causes of mental disorders based on dimensions of observable behavior and neurobiological measures¹⁴.”

- [11] Woo CW, Chang LJ, Lindquist MA, Wager TD. Building better biomarkers: brain models in translational neuroimaging. *Nat Neurosci* **20**, 365-377 (2017).
- [12] Abi-Dargham A, Horga G. The search for imaging biomarkers in psychiatric disorders. *Nat Med* **22**, 1248-1255 (2016).
- [13] Qi S, *et al.* MicroRNA132 associated multimodal neuroimaging patterns in unmedicated major depressive disorder. *Brain*, awx366-awx366 (2018).
- [14] Cuthbert BN. The RDoC framework: facilitating transition from ICD/DSM to dimensional approaches that integrate neuroscience and psychopathology. *World Psychiatry* 2014;13:28-35.

Regarding replication issues, we tested both cross-cohort similarity of brain patterns as well as generalizability of the cognition-predictive models. please also refer to reponse to **Reviewer 1 point #5**, explanation of Fig. 3 as below on page 10 and the section of “Generalization for independent cohorts” on page 18-20 in mansucript.

“...Notably, the anatomic locations of the peak coordinates replicate between the two datasets and include caudate and the bilateral insula for GM, precuneus for fALFF, and PCC/ACC for FA. Specifically, blue arrows indicate the cross-cohort spatial correlations by using the Z thresholded spatial maps of each modality. The cross-cohort correlation between the identified components are GM: $r=0.51$, $p_{perm}=8.0 \times 10^{-4}$, FA: $r=0.59$, $p_{perm}=2.0 \times 10^{-4}$; fALFF: $r=0.45$, $p_{perm}=0.002$ where the significance p_{perm} were resulted from 10000 permutations. Details were provided in Supplementary Table 4....”

In terms of mechanistic clarifications, we have added the following in discussion:

“One major finding was that CEN, SAN and pDMN are three key networks that are particularly important for understanding multiple cognitive functions and their dysfunction in SZ^{53, 54, 55}. These networks are unique in that they can be readily identified across an extremely wide range of cognitive

tasks, and their responses increase and decrease proportionately, with general cognitive task demands⁸. More importantly, they are in accord with the triple network model of major psychopathology⁸. In this model, the SAN, with the anterior insular as its integral causal outflow hub, assists target brain regions in the generation of appropriate behavioral responses to salient stimuli¹⁵. Once such a stimulus or event is detected, the insular facilitates task-related information processing by initiating appropriate transient control signals. These signals engage brain areas that mediate attentional, working memory and higher-order cognitive processes through CEN while disengaging the DMN. Consequently, the aberrant organization of these three core brain networks play a significant role in cognitive impairment of many psychiatric and neurological disorders.

...

our work provides evidence that a synthesis and interaction exist among the three intrinsically coupled networks⁸ in both brain function and structure, which are systematically engaged during cognition and impact multiple cognitive domains.

Within this hierarchy, the SAN would stand in a ‘hub’ position in mediating dynamic interactions between other large-scale brain networks involved in externally oriented attention and internally oriented self-related mental processes⁹. As inferred by Vinod Menon¹⁵, the general consequences for psychopathology are simple: aberrant saliency filtering, detection and mapping result in deviant signaling into and out of the SAN. This in turn has important repercussions for how attentional resources are allocated, and consequently for cognition and behavior. These are suggested as the fundamental mechanisms underlying cognitive dysfunction in psychiatric disorders⁸. Our results also highlight that aberrant functional and anatomical organization within this triple network could be a prominent feature of schizophrenic cognitive impairment. And a proper characterization of these multimodal networks may serve as a neuromarker network to quantify the cognitive dysfunction in psychopathology.”

Reviewer #2 (Remarks to the Author):

In “Searching for multimodal neuromarkers in schizophrenia via cognition guided MRI data fusion: a cross-cohort and multi-domain study,” Sui et al. describe multimodal neuromarkers of cognition in a sample of healthy control samples and patients with schizophrenia. While this study has a good deal to recommend it, including a very interesting supervised data fusion method, there are several aspects of this work which at present diminish its impact.

Thank you for the insightful comments.

1. Style. Despite great interest, I found the paper fairly difficult to follow, which was largely due to a combination of a fairly convoluted analytic (see below) flow and a somewhat cumbersome writing style. For example, the abstract is poorly written, laden with run-on sentences and incomplete information-- “cognitive composite and multiple domain scores were used to guide three-way multimodal MRI data fusion in a discovery cohort (n=294) respectively.”

Thank you for the suggestion. We have rewritten the abstract as following and gone through the whole manuscript carefully to present the analysis flow.

“Imaging neuromarkers that can interrelate cognitive impairments and ultimately predict individual cognitive performance are a major research focus in this new era of psychiatric study. To this end, our goal was to identify multimodal neuromarkers that can be used to quantify and predict cognitive performance, especially impaired in schizophrenia. By supervised learning strategy, multiple cognitive domain scores were used to guide three-way multimodal magnetic resonance imaging (MRI) data fusion in two independent cohorts. Results highlighted the salience network (gray matter, GM), corpus callosum (fractional anisotropy, FA), central executive and default mode networks (fractional amplitude of low frequency fluctuation, fALFF) as potential, modality-specific, neuromarkers of generalized cognition. FALFF features were found to be more sensitive to cognitive domain differences, while the salience network in GM and corpus callosum in FA are highly consistent and predictive on multiple cognitive domains. These modality-specific brain regions define a promising neuromarker network which covaries subject-wisely and has the potential to be used as predictors of multi-domain cognitive scores for new individuals.”

2. Sample and inference. The authors seek to develop a multi-modal dimensional biomarker of cognition. However, this is accomplished in a mixed group of healthy controls and patients with schizophrenia. Because of the large cognitive deficits in patients, the neuromarkers discovered tend to be an admixture of the effects of diagnosis and individual differences in cognition. At one point the authors mention controlling for diagnosis (via partial correlation)—this should be done throughout the paper, which would allow the authors to show that dimensional effects are present in both groups.

Thank you for this suggestion. We have performed the partial correlation analysis throughout the

paper and added the following contents in the main text of the revised manuscript, see page No.13.

“In order to quantify the robustness of the cognition-brain correlation across groups, we performed partial correlation to minimize the group effect. As shown in Supplementary Table 6, in any case of partial correlation, the cognition-imaging correlations remain significant (FDR corrected) after controlling for diagnosis in all four domains.”

Table 6. Partial correlation analysis of FBIRN results after controlling for diagnosis

Modality	GM_IC _{ref}		FA_IC _{ref}		fALFF_IC _{ref}	
	r	p	r	p	r	p
Composite	0.241	9.0e-05*	0.118	0.05	0.275	6.9e-06*
Attention/vigilance	0.137	0.026	0.128	0.037	0.223	2.8e-04*
Working memory	0.291	1.3e-06*	0.188	0.002*	0.265	1.1e-05*
Verbal learning	0.151	0.014	0.156	0.001*	0.139	0.023

3. Methods description. The readability of the manuscript would be improved if some of the relevant methods content could be placed within the context of the results section, especially the Multimodal fusion with reference subsection, to provide intuitions of the MCCAR+jICA throughout the manuscript.

Thank you for the suggestion. We have moved the “Analysis flowchart” section to the “Results” section, and also added a brief description of MCCAR+jICA, see pages 5-6.

4. Parameter tuning. It is difficult to tell, but I imagine regularization is occurring to handle the $p \gg n$ problem of the data used. How regularization is performed, what constraints are used, and how these parameters are tuned are not discussed in the methods. (For example, there is a lambda in one of the equations that is not defined). If parameter tuning is performed, how do the authors ensure that they are not over-fitting the data? If significant tuning is occurring, this method should be cross-validated.

We now clarify that we are indeed cross-validating the parameters. Note, the parameter λ in equation (3) is not used to handle the $p \gg n$ problem of the data but to balance the contribution of cost function: $\|\text{corr}(\mathbf{A}_k, \mathbf{A}_j)\|_2^2$ and $\|\text{corr}(\mathbf{A}_k, \text{ref})\|_2^2$ in equation (3). “MCCA with reference (MCCAR) imposes an additional constraint upon the MCCA (equation (4)) framework to maximize not only the covariations among mixing matrices of each modality, but also the column-wise correlations between \mathbf{A}_k and the reference signal (*ref*).”

$$\max \sum_{k,j=1}^3 \left\{ \|\text{corr}(\mathbf{A}_k, \mathbf{A}_j)\|_2^2 + 2\lambda \cdot \|\text{corr}(\mathbf{A}_k, \text{ref})\|_2^2 \right\} \quad (3)$$

$$\max \sum_{k,j=1}^3 \left\{ \|\text{corr}(\mathbf{A}_k, \mathbf{A}_j)\|_2^2 \right\} \quad (4)$$

“When determining the value of λ , we performed a five-fold cross validation on these 294 subjects for 50 iterations. 4/5 of the data was trained by MCCAR+jICA to be decomposed into \mathbf{A}_{train} and \mathbf{S} , where \mathbf{S} is further used in the remaining 1/5 of testing data to decompose it into \mathbf{A}_{test} and \mathbf{S} . Then we tested the correlation between the reference and the target component of \mathbf{A}_{test} (with the

same IC order of the target component derived from A_{train}) for $5 \times 50 = 250$ times on each modality. As shown in Fig. S1, the mean and standard derivation of correlations of all iterations for the three modalities were calculated and λ was set to the value at which the correlation between target IC and the reference reaches its maximum value ($\lambda = 0.5$ for the FBIRN data). For UNM data, we adopted the same strategy to independently determine the value of λ .”

Figure S1. Correlation of the identified components and CMINDS composite scores across multiple cross-validations. When λ is 0.5, the mean correlation between estimated target IC and composite cognitive scores of all modalities reaches its maximum value. The black line, yellow patch and blue line represent mean, standard error of the mean (SEM) and the standard deviation (SD) of correlations between target IC and composite scores.

We have added the above text and results to the Supplementary “Lamda determination in MCCAR+jICA” section.

5. Significance testing. Relatedly, it is unclear what the null distribution of this method is. If significant regularization is occurring, it is quite possible that the data is over-fit and the null distribution is not centered at zero. Standard permutation testing techniques that show the null, and how the observed correlations compare to the null, would increase confidence in the results substantially.

In response to the reviewer, “we also performed standard permutation test for the correlations listed in the Results. We do this by randomly shuffling Y (cognitive scores) across participants and re-running the correlation analysis (between X [loadings of IC_{ref}] and Y) 10000 times in order to obtain an empirical null distribution. We also record the number of times a correlation coefficient between X and Y exceeds the obtained sample correlation ($r=0.262$, here we take the FA component as an example). Significance cutoffs were determined using the above permutation test (10000 permutations; a cutoff was chosen for a significant alpha of 0.05, that is, 5% of permutations showed one or more significant IC_{ref} vs cognition relationships). As in our results (Fig. 1b), the observed correlation between FA IC_{ref} and cognitive scores obtained on the original data was 0.262, while the sampling distribution of r under randomization is symmetric around 0.0 (Fig. S2), and 20 of the 10000 randomizations exceeded $+0.262$. This analysis quantifies the probability $p=0.002$ of obtaining a particular $r=0.262$ between loadings of FA IC_{ref} and composite cognitive scores by chance.”

Figure S2. Permutation test for the correlation analysis between FA IC_{ref} and cognitive scores (10000 times). The black dotted line indicates ± 0.262 .

“Based on the above permutation procedure, we tested all the correlations for both FBIRN and UNM. FBIRN: $p_{\text{permutation}} = 1.3 \times 10^{-4}$, 0.002, 1.0×10^{-4} for sMRI, dMRI and fMRI, respectively (Fig. 1b). UNM: $p_{\text{permutation}} = 0.02$, 0.01, 0.001 for sMRI, dMRI and fMRI, respectively (Fig. 2b).”

Fig. 1a-b

FBIRN: $p_{\text{permutation}} = 1.3 \times 10^{-4}$, 0.002, 1.0×10^{-4} for sMRI, dMRI and fMRI.

Fig. 2a-b

UNM: $p_{\text{permutation}} = 0.02$, 0.01, 0.001 for sMRI, dMRI and fMRI.

We have added the above contents to the Supplementary “Permutation test” section.

6. Use of replication sample. It was somewhat curious how the authors used the replication sample. Rather than simply applying the model found from the discovery sample, new features were found de-novo. How well does the model from the discovery sample predict general cognition in the replication sample?

Thanks for making this point. We now clarify that we tested replicability in two ways.

We actually used both approaches mentioned by the reviewer. One included replication of the entire analysis pipeline on two independent data sets and comparison of the similarity of the spatial maps and the correlation with cognitive domains. This is a stronger replication criteria as we are allowing the estimation of the components to be data-driven in two independent datasets. The fact that they were highly similar and significantly replicate provides additional support for our proposed approach.

The second approach included regression of the first results onto the second to evaluate the links between the identified spatial networks and cognition. In this case the regions are fixed ahead of time based on the reference data set. Results also replicated in this case. Both cases showed significant replication results, providing strong support for both the proposed approach and the generalizability of the results.

7. Statistical testing of features in replication sample. One of the ways in which the authors assess the replication sample is through spatial correlations of the loading maps. They report high correlations, which are reasonably persuasive. However, statistical tests are also reported, which are uniformly so low as to be below the precision of the test ($p < 10^{-20}$, etc). These values are meaningless, as they seem to consider voxels as “samples” in the correlation. Thus, by simply increasing the resolution of the images in the spatial correlation, the authors are gaining additional “statistical power”, which is arbitrary and based purely on re-sampling. To test these relationships appropriately, a spatial permutation procedure that preserves the structure of the data is necessary—this has frequently been done on a spherical surface (see Gordon et al., Cerebral Cortex 2014 for example).

Thank you for the comments. As suggested, “we now calculated the spatial correlation of the identified target component between two cohorts with only voxels masked at $|Z| > T$. First, the spatial maps were transformed into Z scores and masked at $|Z| > 2$. Then we obtained two masks from FBIRN (mask_FBIRN) and UNM (mask_UNM) respectively, which were used to perform the voxel selection. Only voxels that fell in the union of the masks (mask_FBIRN \cup mask_UNM) were used to calculate the cross-cohort correlation. Thus total number of voxels in calculating the spatial correlation is greatly reduced, e.g., from $n = 153594$ (whole brain voxels) to $m = 1936$ ($T = 2$). Spatial correlation was finally performed on these commonly identified voxels ($m = 1936$) between two cohorts.”

“In addition, we also performed a permutation test to calculate the significance for the cross-

cohort spatial correlation. We do this by randomly shuffling Y (UNM_IC_{ref}) across voxels and re-running the correlation analyses (between X [$FBIRN_IC_{ref}$] and Y) 10000 times in order to obtain an empirical null distribution. We then record the number of times the correlation exceeds the obtained sample correlation. Take GM component for example (Fig. 3), the observed correlation between $FBIRN_IC_{ref}$ and UNM_IC_{ref} was 0.51, while 8 of the 10000 permutations obtained correlations falling out the range of $[-0.51, 0.51]$, thus the probability of $p=8.0 \times 10^{-4}$ was estimated for cross-cohort correlation of $r=0.51$ between GM maps by chance.”

We have revised Fig. 3 by changing p values for spatial correlation to permuted p values and added the above contents to the Supplementary “Spatial correlation” section.

Figure 3. Similarity and covarying patterns of the identified neuromarker networks between cohorts. (a) The identified neuromarker network associated with CMINDS composite in FBIRN cohort. (b) The identified neuromarker network associated with MCCB composite in UNM cohort. The spatial maps were visualized at $|Z|>2$; The blue arrows represent intra-modality similarity (spatial correlation) cross cohorts (the listed p values are permuted p values); the orange arrows represent inter-modality covariation across subjects; and the pink arrows denote the correlations between cognitive composite (the reference) and the identified $IC_{ref_composite}$.

8. Feature overlap. The rationale for finding overlapping features across cognitive sub-domains and then testing them separately was very unclear. Why not simply use the original features defined by the data fusion technique?

Note that one aim of this paper is to extract multimodal features that are significantly associated with cognitive dysfunction. The domain-common multimodal features are the core brain networks that affect across different cognitive domains. Hence, we performed the overlapping procedure across sub-domains to extract common core multimodal co-varying brain networks that can predict multiple cognitive domains for both FBIRN and UNM cohort.

In addition, the global composite is significantly correlated with all other sub-domain scores (Supplementary Table 10, 12). A variety of statistical analyses showed that individual test scores and domain specific scores were significantly inter-correlated and that domain specific scores were highly correlated with a global composite score². Please also refer to **Reviewer 1 point #2**. Certainly, domain-specific research can also be investigated in future study.

9. Resting state pre-processing. The authors used a de-noising pipeline which certainly leaves a huge amount of noise related to motion and physiological artifact in the data, but do not control for motion in the group level analyses using fALFF based components. This of course can be a substantial confound in group level analyses—see for example recent data regarding this risk from the HCP below. Please display the group level correlation between fALFF neuromarkers and mean FD, cognition and FD, and control for FD in group level analyses.

Siegel, J.S., Mitra, A., Laumann, T.O., Seitzman, B.A., Raichle, M., Corbetta, M., and Snyder, A.Z. (2016). Data Quality Influences Observed Links Between Functional Connectivity and Behavior. *Cereb Cortex*.

We are sorry that the section of fMRI preprocessing was not clear enough (we had initially shortened it due to the word limitation). We did remove outlier subjects who have framewise displacements (FD) exceeding 1.0 mm, as well as head motion exceeding 2.0 mm of maximal translation (in any direction of x, y or z) or 1.0° of maximal rotation throughout the course of scanning. We also despiked the fMRI data, and regressed out six head motion parameters, white matter, and cerebrospinal fluid. Results indicate all FDs (mean framewise displacements, mean of root of mean square frame-to-frame head motions assuming 50 mm head radius¹⁶) for all subjects were <0.3 mm at every time point. Note also there is no significant difference between patients and controls on mean FDs, namely,

UNM, HC: mean=0.22±0.12mm, SZ: 0.21±0.11 mm, two sample t-test: $p = 0.77$

FBIRN, HC: mean=0.25±0.18mm, SZ: 0.27±0.21mm, two sample t-test: $p = 0.65$

In response to the reviewer, “we also performed correlation analysis between cognitive scores and mean FDs for both FBIRN and UNM cohort, as displayed in Table 13 and Table 14, none of these tests was significant.”

Table 13. p values for the correlations between mean FD and cognition for FBIRN

	Composite	Speed of Processing	Attention	Working Memory	Verbal Learning	Visual Learning	Reasoning
p value	0.650	0.602	0.450	0.365	0.782	0.685	0.562

Table 14. p values for the correlations between mean FD and cognition for UNM

	Composite	Speed of Processing	Attention	Working Memory	Verbal Learning	Visual Learning	Reasoning
p value	0.139	0.166	0.096	0.112	0.153	0.371	0.637

“Here, we also performed partial correlation analysis for IC_{ref} and cognitive scores by regressing out mean FD, as shown in Table 15 and Table 16, since partial correlation has been proposed as an alternative approach for removing spurious shared variance in correlation analysis¹⁷. It is clear that the correlations between components and cognitive scores are still significant after regressing out FD.”

Table 15. Partial correlation after regressing out mean FD for FBIRN results

Modality	GM_IC _{ref}		FA_IC _{ref}		fALFF_IC _{ref}	
	r	p	r	p	r	p
Composite	0.431	3.5e-11*	0.223	0.028	0.363	1.6e-07*
Attention/vigilance	0.318	1.4e-05*	0.233	0.001*	0.202	0.001*
Working memory	0.290	1.5e-04*	0.183	0.002*	0.232	0.0013*
Verbal learning	0.285	2.2e-04*	0.211	0.005	0.259	1.7e-03*

Table 16. Partial correlation after regressing out mean FD for UNM results

Modality	GM_IC _{ref}		FA_IC _{ref}		fALFF_IC _{ref}	
	r	p	r	p	r	p
composite	0.276	0.014	0.300	0.007*	0.304	0.006*

“For FBIRN cohort, as for the associations between imaging features and mean FD, there are no imaging voxels showed a significant correlation with mean FD after FDR multiple comparison correction ($p_{\text{uncorrected}} < 1.0e-04$) for fALFF, FA and GM. And the correlations between mean FD and fALFF neuromarkers (fALFF_PFC, fALFF_pDMN, shown in Fig. 7a) are not significant either ($p=0.78, 0.56$ for fALFF_PFC and fALFF_pDMN respectively). In UNM cohort, no imaging voxels showed a significant correlation with mean FD after the FDR multiple comparison correction ($p_{\text{uncorrected}} < 0.001$) for any of the three modalities.”

“Finally, the fusion analysis was conducted on the fALFF spatial maps but not the functional connectivity (FC) features as discussed in HCP paper¹⁸. Considering there is no group difference in head motion, and no significant correlations between mean FD and cognitive scores, and partial correlations between IC_{ref} and cognitive scores are still significant after regressing out mean FD, we believe that micro-motion is not a major factor affecting the current results.”

In response to the reviewer, we now have added these details to the Supplementary “Head motion”

and “Data preprocessing” sections as below.

The fMRI data were preprocessed using the automated analysis pipeline¹⁹, whose steps are conducted in SPM8 (<http://www.fil.ion.ucl.ac.uk/spm>) as follows: motion correction to the first image using INRIalign; slice timing corrected to the middle slice; and normalization to MNI space, including reslicing to $3 \times 3 \times 3$ mm voxels. “We removed subjects who have framewise displacements (FD) exceeding 1.0 mm, and head motion exceeding 2.0 mm of maximal translation (in any direction of x, y or z) or 1.0° of maximal rotation throughout the course of scanning. We also despiked the fMRI data, regressed out six head motion parameters, white matter, and cerebrospinal fluid in the denoising procedure. Results indicate FD (mean framewise displacements, mean of root of mean square frame-to-frame head motions assuming 50 mm head radius¹⁶) for all subjects were <0.3 mm at every time point. There is no significant difference between patients and controls on mean FDs; namely, UNM, HC: mean=0.22 ± 0.12mm, SZ: 0.21 ± 0.11 mm, two sample t-test: $p = 0.77$, FBIRN, HC: mean=0.25±0.18mm, SZ: 0.27±0.21mm, two sample t-test: $p = 0.65$. No imaging voxels showed a significant correlation with mean FD after the FDR multiple comparison correction for any of the three modalities, and also no significant correlations between mean FD and cognition. Detailed head motion correction could be found in the Supplementary “Head motion” section.” Data were then spatially smoothed with an 8 mm, full-width half-maximum (FWHM) Gaussian filter. To calculate fractional amplitude of low frequency fluctuations (fALFF)²⁰, the sum of the amplitude values in the 0.01 to 0.08Hz low-frequency power range was divided by the sum of the amplitudes over the entire detectable power spectrum (range: 0–0.25Hz)²¹. “Finally, the fusion analysis was conducted on the spatial maps of fALFF. Considering there is no group difference in head motion, and no significant correlations between mean FD and cognitive scores, and partial correlations between IC_{ref} and cognitive scores still significant after regressing out mean FD, we believe that micro-motion is not a major factor affecting the current results.”

Minor

1. Typo/Grammar Page 17. “Except for” should be “in addition to”, or “besides”.

We have changed “Except for” to “In addition to” in the revised manuscript, see page No.19.

2. Clarify in figures Figure 7 d-e. No labeling of the panels were provided to indicated that these data were from FBIRN alone, not from UNM or combined dataset.

Thank you for the suggestion. We have clarified the dataset cohort in revised Fig. 7d-e, see page No. 17.

Figure 7. Selected multimodal neuromarkers and its predictability on cognitive scores. (a) Four selected MRI neuromarkers from FBIRN data that were used as regressors to predict individual cognitive scores. (b) Prediction of CMINDS composite based on linear regression of the four regressors (mean ROI values in (a)). A correlation of $r = 0.463$ was achieved between the estimated CMINDS composite scores and its true values. (c) By extracting the same four MRI neuromarkers in the UNM cohort via masks and applying the same prediction model as trained in the FBIRN cohort, the MCCB composite scores were predicted for UNM data. A spearman correlation of $r = 0.231$ was achieved between the estimated MCCB composite scores and its true values for 41 HCs (red dots) and 37 SZs (blue dots), indicating excellent generalizability of the proposed prediction model. (d-e) Prediction results for subgroups (HC and SZ only in FBIRN). Either is significant, suggesting the prediction model works for both healthy and diseased people. The gray regions in (b-e) indicate a 95% confidence interval.

Reviewer #3 (Remarks to the Author):

In this manuscript, Sui and colleagues tested whether multimodal MRI data can be leveraged to provide useful predictors of cognitive ability in healthy controls and cognitive impairment in schizophrenia. This topic is of keen interest to the brain imaging community, and this study is noteworthy for its use of multimodal data and cross-validation in an independent dataset. The impact of this manuscript hinges on the generalizability of the identified multimodal biomarkers. On this, my enthusiasm is tempered by the following concerns.

Thank you for the insightful and valuable comments.

Main points:

1. The authors have not given sufficient consideration for head motion as a factor that affects image quality in all three MR modalities included. The influence of head motion needs to be addressed, e.g. by accounting for differences in motion between individuals and particularly between patient and control groups. First, more aggressive removal of motion signals could have been applied to the data (e.g. by regression of whole-brain signal). Second, summary motion parameters should be included in the multiple regression as a covariate. Third, a useful proof that motion does not determine the results might come from using the motion parameters as the reference in the “MCCAR+jICA” clustering and repeating the analyses. Individual differences in head motion could be a uniting factor that links the observed imaging features in the different modalities.

We are sorry that the section of fMRI preprocessing was not clear enough (we had initially shortened it due to the word limitation). “We did remove outlier subjects who have framewise displacements (FD) exceeding 1.0 mm, as well as head motion exceeding 2.0 mm of maximal translation (in any direction of x, y or z) or 1.0° of maximal rotation throughout the course of scanning. We also despiked the fMRI data, and regressed out six head motion parameters, white matter, and cerebrospinal fluid. Results indicate all FDs (mean framewise displacements, mean of root of mean square frame-to-frame head motions assuming 50 mm head radius¹⁶) for all subjects were <0.3 mm at every time point. Note also there is no significant difference between patients and controls on mean FDs, namely,

UNM, HC: mean=0.22±0.12mm, SZ: 0.21±0.11 mm, two sample t-test: $p = 0.77$

FBIRN, HC: mean=0.25±0.18mm, SZ: 0.27±0.21mm, two sample t-test: $p = 0.65$ ”

“Partial correlation has been proposed as an alternative approach for removing spurious shared variance in correlation analysis¹⁷. Here, for the second point, we also performed partial correlation analysis for IC_{ref} and cognitive scores by regressing out mean FD, as shown in Table 15 and Table 16, none of them is significant.”

Table 15. Partial correlation after regressing out mean FD for FBIRN results

Modality	GM_IC _{ref}		FA_IC _{ref}		fALFF_IC _{ref}	
	r	p	r	p	r	p
Composite	0.431	3.5e-11*	0.223	0.028	0.363	1.6e-07*
Attention/vigilance	0.318	1.4e-05*	0.233	0.0014	0.202	0.001*
Working memory	0.290	1.5e-04*	0.183	0.002*	0.232	0.0013*
Verbal learning	0.285	2.2e-04*	0.211	0.005	0.259	1.7e-03*

Table 16. Partial correlation analysis after regressing out mean FD for UNM results

Modality	GM_IC _{ref}		FA_IC _{ref}		fALFF_IC _{ref}	
	r	p	r	p	r	p
composite	0.276	0.014	0.300	0.007*	0.304	0.006*

Figure S4. The identified joint components that are significantly correlated with mean FD. (a) The spatial maps. **(b)** Correlations between loadings of component and mean FD (HC: the red dots, SZ: the blue dots). **(c)** There is no group difference for the loadings of components. The gray regions in (b) indicate a 95% confidence interval.

“The proposed supervised fusion method, MCCAR+jICA²², is usually used to investigate multimodal brain covarying patterns associated with clinical measures of interest, such as symptom severity, intelligence quotient (IQ), cognitive and behavioral measures, or even epigenetic variant (e.g., a microRNA expression¹³).” In response to the reviewer, “we also performed supervised fusion analysis using mean FD as reference. Results (Fig. S4) show that FD associated patterns, which are mainly artifacts in each modality, such as white matter in fALFF and GM, CSF in GM and FA.”

“Finally, the fusion analysis was conducted on the spatial maps of fALFF. Considering there is no group difference in head motion, and no significant correlations between mean FD and cognitive

scores, and partial correlations between IC_{ref} and cognitive scores still significant after regressing out mean FD, we believe that micro-motion is not a major factor affecting the current results.”

2. It would be interesting to see a null model – for instance using a permuted vector as reference – to give us an indication of what kinds of spatial patterns we might expect by chance. This would give us confidence that the observed patterns are specific to the cognitive measures being used as priors. I found it interesting that the areas identified in the fMRI and FA data were regions that typically show the highest signal in each of those modalities. i.e. the corpus callosum and corticospinal tracts typically show high FA, and the default mode network typically exhibits high signal in fMRI. Is it possible that these regions were clustered together due to their SNR properties, rather than any interesting associations with cognitive measures?

In response to the reviewer, “we permuted the reference vector (cognitive scores) in the supervised fusion analysis. The goal is to compute the null model of spatial patterns that are observed by chance. To do this we hold imaging variables (e.g. [X_1 , X_2 , X_3]) constant, and permute the reference (global cognitive scores) against them. Thus each X_i is randomly paired with a reference. This permuted reference was then used as reference in a supervised fusion analysis (MCCAR+jICA). By repeating this process a large number of times (1000), we obtain 1000 three-MRI covarying patterns associated with the permuted reference. We also record the number of times each spatial pattern occurs. Here we presented the most frequently occurring voxels (those which occur more than 60% of the time) associated with the permuted cognitive scores, as shown in Fig. S5b. Note that the permuted null model of spatial patterns is different from the triple network as identified in Fig. 3-4, confirming that the observed pattern as presented in our results, including GM_SAN, FA_CC, fALFF_PFC and fALFF_pDMN are specific to the cognitive measures but not a random null pattern.”

Figure S5. (a) The covarying pattern for the original cognitive scores. (b) The most frequently occurring (voxels with more than 60% occurrences) covarying pattern associated with 1000 times permuted cognitive scores.

The identified original multimodal brain networks also show significant correlation with all CMINDS cognitive domain scores, as listed in Supplementary Table 7. In addition, two-sample t-tests between HC and SZ show significant group differences for GM_SAN ($p=6.8 \times 10^{-9*}$), FA_CC ($p=0.001$) and fALFF_PFC ($p=7.0 \times 10^{-9*}$) respectively. The effect size for separating groups was also computed with Cohen's $d = 0.698, 0.487, 0.623$ for GM_SAN, FA_CC and fALFF_PFC

respectively. More importantly, the multimodal features we identified, can predict individualized cognition for independent cohort and for multiple cognitive domains. While the null patterns are not correlated with CMINDS cognitive domains scores nor group discriminative between patients and controls, neither predictive.

We have added the above contents in the Supplementary “Validating the specificity of the nueromarker to cognition” section.

3. I found the generalization of the derived model to be underwhelming. Evidence for generalization came from two forms, 1) the spatial similarity of the IC maps defined separately in the two datasets, and 2) application of the regression model derived from one dataset to the other:

3.1 Spatial similarity: The spatial maps derived from the two datasets were correlated to ~ 0.5 , however the details of how the spatial correlation was performed are not fully explained. Were the spatial maps masked to exclude white matter voxels? For instance, in the GM correlation in Fig 3, were all voxels included in the final correlation analysis, or was the correlation restricted to the GM voxels? If the former, this would inflate the correlations. More information about the spatial correlation would be welcome.

Thank you for the comments. We also modified our testing approach to incorporate a permutation analysis. “Here, take GM as an example. We now calculated the spatial correlation of the identified target component between two cohorts with only voxels masked at $|Z| > T$ (threshold). First, the spatial maps were transformed into Z scores and masked at $|Z| > 2$. Then we obtained two masks from FBIRN (mask_FBIRN) and UNM (mask_UNM) respectively, which were used to perform the voxel selection. Only voxels that fell in the union of the masks (mask_FBIRN \cup mask_UNM) were used to calculate the cross-cohort correlation. Thus total number of voxels in calculating the spatial correlation is greatly reduced, e.g., from $n=153594$ (whole brain voxels) to $m=1936$ ($T=2$). Spatial correlation was finally performed on these commonly identified voxels ($m=1936$) between two cohorts.”

“In addition, we also performed a permutation test to calculate the significance for the cross-cohort spatial correlation. We do this by randomly shuffling Y (UNM IC_{ref}) across voxels and re-running the correlation analyses (between X [FBIRN IC_{ref}] and Y) 10000 times in order to obtain an empirical null distribution. We then record the number of times the correlation exceeds the obtained sample correlation. Take GM component for example (Fig. 3), the observed correlation between FBIRN IC_{ref} and UNM IC_{ref} was 0.51, while 8 of the 10000 permutations obtained correlations falling out the range of $[-0.51, 0.51]$, thus the probability of $p=8.0 \times 10^{-4}$ was estimated for cross-cohort correlation of $r=0.51$ between GM maps by chance.”

We have updated Fig. 3 by replacing p values with the permuted p , and we have added the above contents in Supplementary “Spatial correlation” section.

3.2 Regression model: The high p values in Figs 1 and 7 and Table 1 suggest that there is a large degree of overfitting happening in the training dataset. Hence the frequent reporting of p -values from the training dataset is misleading as the real test of the model's validity is its generalizability to the replication dataset (i.e. Figure 7c). Here, the model fit is not very good ($r=0.2$, $p=0.04$), and to my eye there is no clear relationship between True and Predicted values. It looks to me like the slope is influenced by the red data point at the top right of the plot, and I would need to see the same plot with outliers removed before I could be convinced of any relationship.

In response to the reviewer, we calculated the correlation after removing the red point, as shown in the right figure. After removing the red point, the correlation between the true and the predicted values remains significant ($r=0.209$, $p=0.044$). Note that the red data point at the top right is not an outlier, but is very close to the diagonal line, though looks different from others.

Moreover, we performed permutation test for the correlations between the predicted value and true value of Fig. 7c. By randomly shuffling Y (the predicted cognitive scores) across participants estimating correlations (between X [true values] and Y) 1000 times, we recorded the number of times a correlation coefficient between X and Y exceeding the original correlation ($r=0.231$). Significance cutoffs were determined as shown in Fig. S6. Namely, 32 of the 1000 randomizations exceeded $r=0.231$. This analysis quantifies the probability $p=0.032$ of obtaining a particular $r=0.231$ between true value and predicted value by chance.

Furthermore, the model generalization from CMINDS to MCCB work well not only for composite score, but also for most cognitive domains (see Table 1 below), implicating the appreciable generalizability of the cognition-predictive models. Reminder that CMINDS and MCCB are two similar but not identical cognitive measurement systems¹⁰; therefore, the cross-cohort generalization is a powerful evidence to validate the predictability of the identified neuromarkers.”

Figure S6. Permutation test for the correlation between true value and predicted cognitive scores of UNM cohort (1000 times). The black dotted line indicates ± 0.231 .

Table 1. Prediction of CMINDS domains using the 4 neuromarkers and their generalization to MCCB domains

Predicted measures	CMINDS (FBIRN)		MCCB* (UNM)	
	r	p	r	p
Cognitive composite	0.463	2.8e-15	0.231	0.04
Speed of processing	0.470	4.8e-16	0.206	0.05
Attention/vigilance	0.332	3.5e-08	0.231	0.038
Working memory	0.402	8.0e-12	0.218	0.05
Verbal learning	0.371	3.8e-10	0.230	0.04
Reasoning/problem solving	0.330	3.5e-08	0.193	0.08

*Prediction of MCCB domain scores using the same model trained for corresponding CMINDS domain scores in FBIRN cohort.

We have added more explanation of the above details in manuscript, see page 19.

... “The model generalization from CMINDS to MCCB work well not only for composite score, but also for most cognitive domains (see Table 1 below), implicating the appreciable generalizability of the cognition-predictive models. Reminder that CMINDS and MCCB are two similar but not identical cognitive measurement systems¹⁰; therefore, the cross-cohort generalization is a powerful evidence to validate the predictability of the identified neuromarkers. As shown in Fig. 7c, a correlation of $r = 0.231$ was achieved between estimated MCCB composite scores and its true values after performing 1000 permutations, resulting in the probability of $p=0.032$, validating the excellent robustness of the proposed prediction model.

4. On a similar note, if the composite cognitive scores were used as a reference for the definition of the components, how informative is it to report a p value for the correlation between component loadings and composite cognition scores in Figure 1? Is it surprising that there is a good fit of the loadings to the scores?

As discussed in our original paper, not all components obtained from MCCAR+jICA²² are correlated with the reference. Usually, there will be one or more component identified to be correlated with the referred signal. In our case, 24 components were derived for each modality in FBIRN data, among which one joint component shows significant correlations with the referred CMINDS cognitive composite. Therefore, it is not surprising to discover a good fit of the loadings of one IC to the cognitive score, since our goal is to identify the most correlated brain components. However, the IC_{ref} is selected only when it is both correlated with the reference and group discriminative between schizophrenia and healthy controls.

5. As a general point, I found the motivation of the study slightly muddled. The stated aim is to unearth markers of cognitive ability, however the authors also study cognitive impairment in schizophrenia. Why are these two aims carried out in the same datasets? Does the inclusion of schizophrenic patients

determine the results? And if so, doesn't this make the identified markers less generalizable indicators of normal cognitive ability? Although the authors did test that the model holds when patients were excluded, the model was still derived using both patients and controls.

Thanks for this suggestion. Note that the aim and topic of this paper is to extract multimodal features that are associated with cognition as well as group discriminating between HC and SZ, that is, we are interested in identifying cognitive biomarkers that may also separating groups. This is consistent with the scope of the research domain criteria (RDoC) project proposed by NIMH, which aims to identifying dimensional measures that may help separate or redefine patient subtypes and controls¹⁴. If we analyze each group separately and then compare, this will bias the results as the algorithm will be given information about the group membership. Indeed, this is one of the major advantages of computing the C_k and A_k from a joint decomposition. Another approach is to train the model to identify networks on a separate healthy dataset and then identify specific predictive networks using independent patient and control data. As this would require yet more patient/control data, and would represent a substantially different approach, we plan to explore this in future work. We now mention this in the revised manuscript.

Other points:

6. Abstract:

6.1 The term “neuromarker” is defined in the introduction, after being mentioned in the abstract.

As the abstract word limit is 150 words, we initially defined the term “neuromarker” in the introduction section but not in the abstract. We have now revised the abstract as suggested, see page 3.

6.2 The inclusion of schizophrenic patients in the dataset is not mentioned in the abstract.

Thank you for the suggestion. We have revised the abstract as follows.

“Our ultimate goal was to identify multimodal neuromarkers that can be used to predict cognitive performance in schizophrenia patients using multivariate data mining.”

7. Introduction:

7.1 P3 > Definition of “neuromarker”: The term “neurocognitive” seems misplaced here.

Thank you for the suggestion. We have changed the definition of “neuromarker” as follows.

“Here a “neuromarker” is defined as a brain measure that is associated with a cognitive or behavioral outcome and which is predictive of individual performance²³.”

7.2 P3 > Grammar: ‘For example, Rosenberg et al., demonstrate that a set of whole-brain functional network strength’

Thank you for pointing this out. We have changed to “Rosenberg *et al.* demonstrated that whole-brain functional connectivity strength may serve as a neuromarker of sustained attention for both healthy and disease assessments.” in the new revised paper, see page No. 3.

7.3 P3 > “Cognitive Composite ability” – Please explain how this composite measure is defined.

“The CMINDS-based¹⁰ cognitive domains, based on comparable tests to those assessed by the MCCB, were as follows: (1) *Speed of Processing*. This domain score was based on the mean of (a) the log-transformed, negated (worse performance is lower) elapsed time (in seconds) during *Trails A*, (b) the number of correct in set responses in 60 seconds on trial 1 of the *Category Fluency Test – Animals*, and (c) the number of correct responses during the *Symbol Digit Association Test* z-scores; (2) *Attention/Vigilance*. This domain score was based on the *d*-prime across blocks A–C of the *Continuous Performance Test* z -scores; (3) *Working Memory*. This domain score was based on the mean of (a) the sum of the number of correct on the *Visual Spatial Sequencing Test – Forward and Backward* condition, and (b) the total correct on the *Letter Number Span* z -scores; (4) *Verbal Learning*. This domain score was based on the total number of correctly recalled target words for all three trials on the *Semantic Verbal Learning Test* z-scores; (5) *Visual Learning*. This domain score was based on the square-transformed total of the *Visual Figure Learning Test* z-scores, and (6) *Reasoning/Problem Solving*. This domain score was based on the square transformed *Maze Solving Test* total score z-scores. Finally, the CMINDS Composite Score was defined as the mean of all six normalized domain scores.”

We have added the above contents to the Supplementary “CMINDS scores” section.

7.4 Aim 1 – Given the multimodal nature of the analysis, how do the authors know a priori that the “identified multimodal neuromarker” will be a ‘network’, classically defined based on anatomical connectivity, or spontaneous correlations between brain regions, or perhaps even distributed regions showing similar task activation profiles? The authors need to define what they mean by terms like ‘network’ and ‘neuromarker’ more carefully. For instance, in the Results section the term “multimodal neuromarker network” is introduced. Are these ‘networks’ only definable by a confluence of imaging techniques? How does this then relate to the “brain networks” as defined classically?

Sorry for the lack of clarity. Based on the triple network theory⁸, we hypothesized that the salience network (SAN), central executive network (CEN), and default mode network (DMN) would play pivotal roles in cognitive deficits. Here a “neuromarker” is defined as a brain measure that is associated with a cognitive or behavioral outcome and can further predict individual performance²³. While the “network” is defined as a map whose regions exhibit similar subject-wise covariation, which is the same definition of network as laid out in²⁴. So, in our paper, the term “multimodal neuromarker network” represents multimodal covarying and modality-specific brain regions that can jointly predict cognitive performance for new individuals. We now have added such a definition in introduction.

“Aberrant organization and functioning of the salience network (SAN), central executive network (CEN), and default mode network (DMN) are prominent features of several major psychiatric and neurological disorders and are also fundamental mechanisms underlying cognitive dysfunction in many psychiatric disorders²⁵. Based on the triple network theory⁸, we hypothesized that SAN, CEN

and DNM would play pivotal roles in cognitive deficit in schizophrenia²⁶, which may consist a multimodal neuromarker network that is defined as modality-specific brain regions exhibiting similar subject-wise covariation that can jointly predict cognitive performance for unseen individuals.”

7.5 Aim 2 – Similarly, “What are the neuromarker networks associated with specific representative cognitive domains?” > By the authors definition aren’t the “neuromarker networks” defined by their association with cognitive domains? Perhaps remove the word “neuromarker”?

Thank you for the suggestion. We have removed the word “neuromarker” and changed to “What are the brain networks associated with specific representative cognitive domains?” in Aim 2. Our goal is to investigate the association between the identified multimodal brain networks and multiple cognitive domains (attention, working memory and verbal learning).

7.6 Aim 3 – What is a “cognitive domain discrepancy”?

We now rephrased the word in revised manuscript. “cognitive domain discrepancy” refers to brain regions that differentiate between cognitive domains (i.e., less similarity in Fig. 5e). For example, Fig. 6 illustrates the fALFF components associated with different cognitive domains. Besides the commonly shared prefrontal regions, each domain corresponds to its specific fALFF maps. We now change the corresponding contents in the revised manuscript on page 15-16.

“We found that structural brain measures (GM/FA) express more similarity and are highly consistent across multiple cognitive domains, whereas the fALFF functional measure is more sensitive to differences between cognitive domains (Fig. 5e). With respect to fALFF (Fig. 6), pDMN was identified only for the cognitive composite and working memory domains, while other subcortical and cortical regions such as thalamus, hippocampus, STG and visual cortex occur differently depending on different domains. ...”

Figure 6. Spatial maps of fALFF maps associated with four CMINDS cognitive domains: composite, working memory, attention and verbal learning domain respectively.

Figure 5 Comparison of the multimodal “neuromarker networks” associated with cognitive composite and three domain scores. The spatial maps of the neuromarker network obtained under the guidance of reference (a) composite (red), (b) working memory (green), (c) attention (cyan) and (d) verbal learning (magenta) are displayed in different color mapping. (e) indicates the pair-wise, cross-domain, spatial correlation among brain maps of FBIRN_IC_{ref_composite}, FBIRN_IC_{ref_memory}, FBIRN_IC_{ref_attention}, and FBIRN_IC_{ref_learning} for GM, FA and fALFF respectively. The darker blue and larger shading denotes higher correlation. Note that GM and FA maps demonstrate more consistent patterns across cognitive domains ($r > 0.6$), while fALFF maps exhibit more variance ($r < 0.5$ but still significant). This suggests that a functional measure such as fALFF may differentiate between cognitive domains more sensitively, whereas structural brain patterns (GM/FA) are more consistent across cognitive domains.

7.7 Given that the central hypothesis of the study follows from the Triple Network Theory proposed by Vinod Menon, it would be useful to have a summary of this in the introduction, particularly in regards to the roles of the three networks in question, and schizophrenia.

Thank you for the suggestion. We have added the following contents in introduction in the revised manuscript, see page No. 4.

“...we searched for multimodal neuromarkers that can be used to quantify and predict cognitive performance, especially impaired in schizophrenia, by successive multivariate data mining and model generalization.”

“We hypothesized that the multimodal covarying and modality-specific salience network (SAN), central executive network (CEN), and posterior default mode network (pDMN) would play pivotal roles in cognitive deficits, based on the triple network theory⁸. Aberrant organization and functioning of the CEN, SAN and DMN are prominent features of several major psychiatric and neurological disorders and are also fundamental mechanisms underlying cognitive dysfunction in many psychiatric disorders²⁵. Furthermore, patients with schizophrenia show both structural and functional deficits in all three networks²⁶.”

7.8 P4 > “...each of the four cognitive domains...” > the domains being referred to have not yet been defined.

Sorry for the lack of clarity. We have added the explanation of the four cognitive domains in the new revised paper (page 4). The revised text follows:

“After discovering the “neuromarker network” associated with each of four cognitive domains (Computerized Multiphasic Interactive Neuro-cognitive System [CMINDS]¹⁰ composite, attention, working memory and verbal learning), we compared them in two ways: 1) across domains—to reveal the commonality and uniqueness of the domain-specific “neuromarker networks”; 2) across modalities—to evaluate which imaging modality is more sensitive to cognitive domain.”

7.9 P4 > Grammar: “Remarkably, these models were successful in predicting for new individuals in an independent cohort on the corresponding cognitive metrics. This validates the generalizability of the identified modality” > “Remarkably, these models were successful in predicting the corresponding cognitive metrics for new individuals in an independent cohort”.

Thank you for helping improve this sentence. We have used the suggested text (page 5) and also carefully checked the entire manuscript.

8. Methods:

8.1 P25 > Typo: “we can obtained”

We have made this correction (page 28).

8.2 P25 > Please state explicitly what A_j denotes in equation 3.

According to blind source separation theory²⁷, independent component analysis (ICA) describes how the observed data (X) are generated by a process of mixing (A) the components (C), that is $X = AC$. In our supervised fusion analysis, there are 3 multimodal datasets X_k , and each is a linear mixture of components C_k with a nonsingular mixing matrix A_k , $k = 1,2,3$, denoting the modality. Namely, $X_k = A_k C_k$, where X_k is a subjects-by-voxels feature matrix (fALFF, FA or GM) and A_k is a subjects by number of components (M) mixing matrix.

$$\max \sum_{k,j=1}^3 \left\{ \|\text{corr}(A_k, A_j)\|_2^2 + 2\lambda \cdot \|\text{corr}(A_k, \text{ref})\|_2^2 \right\} \quad (3)$$

So A_k and A_j in equation (3) are mixing matrix for different modalities (fALFF, FA and GM).

8.3 P25 > Please provide more details about how the target IC_{ref} is defined. Is the component that has highest correlation with the ref selected?

IC_{ref} is the selected component both correlated with the reference and group discriminating between patients and controls. Take fALFF_ IC_{ref} _composite as an example, it means that this component of fALFF is significantly correlated with CMINDS composite cognitive scores and group discriminating between schizophrenia and healthy controls. In most cases, IC_{ref} has the highest correlation with reference, but there is still another consideration that we selected group discriminative

component.

8.4 P26 > Typo: “After defined”.

We have made this correction.

8.5 P26 > The section titled: “Predictive neuromarker extraction” was hard to follow. At one point we are referred to Figure 7a for explanation of the methods.

Sorry for the lack of clarity. We have rewritten this section to make it clearer. See manuscript pages No. 28.

“After identifying the neuromarker networks of four CMINDS cognitive domains from the FBIRN data (Fig. 5), we extracted the brain regions which were consistently involved within each modality as potential neuromarkers. Take the feature extraction of GM as an example. After converting component GM_IC_{ref} into Z scores and thresholding at $|Z| \geq 2$, masks of GM_IC_{ref} for each of the four cognitive domains (composite, attention, working memory and verbal learning) were generated. A map of regions included in each of these four GM masks reveals the common ROIs across the four cognitive domains. The final GM mask of GM_SAN is shown in Fig. 7a. This mask of GM was then used to extract ROI features from every subject. The mean of the voxels within the obtained ROI was calculated for each subject, generating a $N_{subj} \times 1$ feature vector for GM_SAN. The other modalities (dMRI and fMRI) were processed in the same way to get the FA_CC and fALFF_PFC feature vectors. The fourth fALFF_pDMN feature was extracted from fALFF_IC_{ref_composite} from the FBIRN data. The resulting regions were those included in the triple network hypothesis⁸. Finally, we formed a feature matrix in dimension of $N_{subj} \times 4$ for the FBIRN data. For the UNM cohort, the ROI features of each modality were extracted by applying the four masks generated from the FBIRN to the UNM data. Following this, the mean of each ROI was calculated for each subject, resulting in a $N_{subj} \times 4$ feature matrix for the UNM cohort.”

9. Figures:

9.1 Fig 1: Typo: “congnition”.

We have made this correction.

Figure 1. The identified joint components that are both significantly correlated with CMINDS composite scores and group-discriminating in all modalities. (a) The spatial maps visualized at $|Z|>2$; the positive Z-values (red regions) means HC>SZ and the negative Z-values (blue regions) means HC<SZ. **(b)** Correlations between loadings of component and CMINDS composite scores (HC: the red dots, SZ: the blue dots); thus SZ corresponds to worse cognitive performance and lower loading weights compared to HC. **(c)** Boxplot of the loading parameters of FBIRN_IC_{ref_composite} that were adjusted as HC>SZ on the mean of loadings for each modality, with the p values of two sample t-tests between HC and SZ shown bottom. The gray regions in **(b)** indicate a 95% confidence interval.

References

1. Woods SW. Chlorpromazine equivalent doses for the newer atypical antipsychotics. *The Journal of clinical psychiatry* **64**, 663-667 (2003).
2. Dickinson D, Harvey PD. Systemic hypotheses for generalized cognitive deficits in schizophrenia: a new take on an old problem. *Schizophrenia bulletin* **35**, 403-414 (2009).
3. White TP, Joseph V, Francis ST, Liddle PF. Aberrant salience network (bilateral insula and anterior cingulate cortex) connectivity during information processing in schizophrenia. *Schizophr Res* **123**, 105-115 (2010).
4. Palaniyappan L, Mallikarjun P, Joseph V, White TP, Liddle PF. Reality distortion is related to the structure of the salience network in schizophrenia. *Psychological medicine* **41**, 1701-1708 (2011).
5. Palaniyappan L, Liddle PF. Does the salience network play a cardinal role in psychosis? An emerging hypothesis of insular dysfunction. *Journal of psychiatry & neuroscience : JPN* **37**, 17-27 (2012).
6. Sridharan D, Levitin DJ, Menon V. A critical role for the right fronto-insular cortex in switching

- between central-executive and default-mode networks. *Proc Natl Acad Sci U S A* **105**, 12569-12574 (2008).
7. Hill SK, *et al.* Neuropsychological impairments in schizophrenia and psychotic bipolar disorder: findings from the Bipolar-Schizophrenia Network on Intermediate Phenotypes (B-SNIP) study. *Am J Psychiatry* **170**, 1275-1284 (2013).
 8. Menon V. Large-scale brain networks and psychopathology: a unifying triple network model. *Trends Cogn Sci* **15**, 483-506 (2011).
 9. Downar J, Blumberger DM, Daskalakis ZJ. The Neural Crossroads of Psychiatric Illness: An Emerging Target for Brain Stimulation. *Trends Cogn Sci* **20**, 107-120 (2016).
 10. van Erp TG, *et al.* Neuropsychological profile in adult schizophrenia measured with the CMINDS. *Psychiatry Res* **230**, 826-834 (2015).
 11. Woo CW, Chang LJ, Lindquist MA, Wager TD. Building better biomarkers: brain models in translational neuroimaging. *Nat Neurosci* **20**, 365-377 (2017).
 12. Abi-Dargham A, Horga G. The search for imaging biomarkers in psychiatric disorders. *Nat Med* **22**, 1248-1255 (2016).
 13. Qi S, *et al.* MicroRNA132 associated multimodal neuroimaging patterns in unmedicated major depressive disorder. *Brain*, awx366-awx366 (2018).
 14. Cuthbert BN. The RDoC framework: facilitating transition from ICD/DSM to dimensional approaches that integrate neuroscience and psychopathology. *World Psychiatry* **13**, 28-35 (2014).
 15. Menon V, Uddin LQ. Saliency, switching, attention and control: a network model of insula function. *Brain structure & function* **214**, 655-667 (2010).
 16. Allen EA, *et al.* A baseline for the multivariate comparison of resting-state networks. *Front Syst Neurosci* **5**, 2 (2011).
 17. Smith SM, *et al.* Network modelling methods for FMRI. *Neuroimage* **54**, 875-891 (2011).
 18. Siegel JS, *et al.* Data Quality Influences Observed Links Between Functional Connectivity and Behavior. *Cerebral cortex* **27**, 4492-4502 (2017).
 19. Bockholt HJ, *et al.* Mining the mind research network: a novel framework for exploring large scale, heterogeneous translational neuroscience research data sources. *Front Neuroinform* **3**, 36 (2010).
 20. Zou QH, *et al.* An improved approach to detection of amplitude of low-frequency fluctuation (ALFF) for resting-state fMRI: fractional ALFF. *J Neurosci Methods* **172**, 137-141 (2008).
 21. Turner JA, *et al.* A multi-site resting state fMRI study on the amplitude of low frequency fluctuations in schizophrenia. *Front Neurosci* **7**, 137 (2013).
 22. Qi S, *et al.* Multimodal Fusion With Reference: Searching for Joint Neuromarkers of Working Memory Deficits in Schizophrenia. *IEEE transactions on medical imaging* **37**, 93-105 (2018).
 23. Gabrieli JD, Ghosh SS, Whitfield-Gabrieli S. Prediction as a humanitarian and pragmatic contribution from human cognitive neuroscience. *Neuron* **85**, 11-26 (2015).
 24. Erhardt EB, Allen EA, Damaraju E, Calhoun VD. On network derivation, classification, and visualization: a response to Habeck and Moeller. *Brain connectivity* **1**, 1-19 (2011).
 25. Menon V, Uddin LQ. Saliency, switching, attention and control: a network model of insula function. *Brain Struct Funct* **214**, 655-667 (2010).
 26. Palaniyappan L, Mallikarjun P, Joseph V, White TP, Liddle PF. Regional contraction of brain surface area involves three large-scale networks in schizophrenia. *Schizophrenia research* **129**, 163-168 (2011).
 27. A. Hyvärinen EO. Independent component analysis: algorithms and applications. *Neural*

Networks, 411–430 (2000).

Reviewers' comments:

Reviewer #1 (Remarks to the Author):

The authors have given a detailed response to my queries. I find that the current version is satisfactorily written to recommend its publication.

Reviewer #2 (Remarks to the Author):

The authors have answered nearly all concerns with an excellent revision. One question remains: if lambda is tuned separately for each fold (and also in the separate test set), does this mean that different models are created in each fold? How did the authors integrate feature weights across folds? Finally, was lambda tuned similarly in the permutation testing for the null model?

Reviewer #3 (Remarks to the Author):

Second review of Sui et al. for Nature Communications

Sui and colleagues have conducted a series of additional analyses to bolster the validity of their multimodal biomarkers that predict cognitive ability and classify schizophrenic and control individuals. Notably they have used permutation testing at several instances to develop null models and have tested for the influence of head motion more rigorously. I also appreciate their efforts to clarify the text, though the language used is still hard to follow in many places.

Although I am sympathetic to the manuscript's aims and technicality, I remain underwhelmed by the key analysis, which is the cross-validation of the model in the independent dataset (Figure 7c). With the outlier excluded, the significance of the model fit is borderline ($p=0.044$, $r=0.209$). The authors may argue that this is impressive in itself (though "indicating excellent generalizability" is going too far), however, given the recent drive for reproducibility in psychology, I would feel much more at ease with this manuscript being published in an NPG journal if the authors were able to reproduce the result in another dataset, e.g. the Human Connectome Project Data (which also has cognitive measures and pre-processed structural and diffusion MR data). Otherwise, unfortunately as it stands my opinion is that this work would be better suited to a more specialized journal. I am unconvinced that the identified 'neuromarkers' will be clinically useful or inform us about brain function and cognitive ability.

Other points -

In the null analysis where cognitive scores were permuted before the supervised fusion analysis, it is not clear why the authors chose to report voxels that appear in more than 60% of the maps, rather than show the actual maps that came out of this analysis. The useful comparison would be to see an example spatial map, e.g. from one of those permutations. Do any of the permuted maps look like the reference maps obtained using the cognitive scores? Do any of the permuted maps look like biologically plausible networks? If so, then to my mind that would indicate serious problems for the analysis.

Also - why does supervised fusion using the permuted scores give us a near perfect delineation of the lateral ventricles? Something odd is happening here, highlighting that inserting junk into the pipeline seems to produce seemingly interesting spatial patterns.

I am concerned that the permutation analysis used to calculate the significance of the cross-cohort spatial correlation does not provide an adequate null model for the spatial correlations. As I've

understood it, the ICref maps were calculated from smooth data which has some degree of spatial autocorrelation between neighboring voxels - due in part to neural signal, but also to non-neural factors (e.g. Kriegeskorte et al., 2008). If the voxels in the ICref map are randomly shuffled, this obliterates all spatial relationships, including these artefactual ones due to spatial proximity. In other words, the permutation null model has no proximity-related correlations whereas the cross-cohort correlations do, and comparing the two is not valid. The problem here is that the autocorrelations will inflate the cross-cohort correlation, even if the ICref map contained no biologically meaningful spatial patterns.

Artifactual time-course correlations in echo-planar fMRI with implications for studies of brain function. Kriegeskorte, N., Bodurka, J., & Bandettini, P. (2008) *International Journal of Imaging Systems and Technology*, 18, 345-349.

Cluster failure: Why fMRI inferences for spatial extent have inflated false-positive rates. Eklund, A., Nichols, T.E., & Knutsson, H. (2016) *PNAS* 113, 7900-7905.

I appreciate the effort to clarify the terms in the manuscript. Some aspects are still confusing though, for instance, the "multimodal neuromarker network" seems to be comprised of three networks, the DMN, SAN and CEN. Is there not a separate term that the authors could use for the identified marker than the word "network"? Such as, a 'signature' or just a 'multimodal marker' or 'imaging correlate' etc?

Similarly, the authors have a very specific definition of the term 'neuromarker': "Here a "neuromarker" is defined as a brain measure that is associated with a cognitive or behavioral outcome and can further predict individual performance". If the authors don't want to define "neuromarker" in the abstract, they should use a different term instead. For instance, the term "biomarker" is commonly used in the field. The first sentence of abstract doesn't make sense without the term 'neuromarker' being defined.

Reviewer #1 (Remarks to the Author):

1.1 The authors have given a detailed response to my queries. I find that the current version is satisfactorily written to recommend its publication.

Thank you for your comments.

Reviewer #2 (Remarks to the Author):

2.1 The authors have answered nearly all concerns with an excellent revision. One question remains: if lambda is tuned separately for each fold (and also in the separate test set), does this mean that different models are created in each fold? How did the authors integrate feature weights across folds? Finally, was lambda tuned similarly in the permutation testing for the null model?

Sorry for the unclearness of the lambda tuning. For each five-fold cross-validation, lambda is the same for all folds, not tuned separately for each fold. The difference between folds is only because the 4/5 training data and the remaining 1/5 test data were different; Therefore, the IC_ref extracted by MCCAR+jICA are slightly different at each fold, which was used for projecting in the test data and obtaining the corresponding loadings. Thus 4/5 of the data was trained by MCCAR+jICA and then decomposed, i.e., $\mathbf{X}_{train} = \mathbf{A}_{train} \times \mathbf{S}_{train}$, where \mathbf{S}_{train} is further used for the remaining 1/5 of testing data to obtain \mathbf{A}_{test} ($\mathbf{A}_{test} = \mathbf{X}_{test} \times pinv(\mathbf{S}_{train})$). The loadings of IC_ref in \mathbf{A}_{test} were then correlated with the reference at each fold, and the mean of their correlations of all folds were compared. As shown in Supplementary Fig. 1, for lambda ranging in [0.1:0.1:1], lambda=0.5 was selected, since the correlation between target IC and the reference is maximum among all choices.

Supplementary Figure 1. When λ is 0.5, the mean correlation between estimated target IC and composite cognitive scores of all modalities reaches its maximum value. The black line, yellow patch and blue line represent mean, standard error of the loading correlations between targeted IC in test data and the composite scores.

Finally, in the permutation testing for the null model, the lambda tuning (five-fold cross validation) was the same as mentioned above. Namely, when we permuted the reference vector (cognitive scores) in the supervised fusion analysis, the goal is to compute the null model of spatial patterns that are observed by chance. To do this, we hold imaging variables (e.g. [$\mathbf{X}_1, \mathbf{X}_2, \mathbf{X}_3$]) constant, and permute the reference (global cognitive scores) against them. Thus, each X_i is randomly paired with a permuted reference and we ran MCCAR+jICA 1000 times (corresponding to 1000 specific reference). For every one of the 1000 permuted references, we performed similar five-fold cross validation (1000 times as in Supplementary Fig. 1) to determine the lambda in each specific MCCAR+jICA.

Reviewer #3 (Remarks to the Author):

Second review of Sui et al. for Nature Communications.

3.1 Sui and colleagues have conducted a series of additional analyses to bolster the validity of their multimodal biomarkers that predict cognitive ability and classify schizophrenic and control individuals. Notably they have used permutation testing at several instances to develop null models and have tested for the influence of head motion more rigorously. I also appreciate their efforts to clarify the text, though the language used is still hard to follow in many places.

Thank you for the comments.

3.2 Although I am sympathetic to the manuscript's aims and technology, I remain underwhelmed by the key analysis, which is the cross-validation of the model in the independent dataset (Figure 7c). With the outlier excluded, the significance of the model fit is borderline ($p=0.044$, $r=0.209$). The authors may argue that this is impressive in itself (though "indicating excellent generalizability" is going too far), however, given the recent drive for reproducibility in psychology, I would feel much more at ease with this manuscript being published in an NPG journal if the authors were able to reproduce the result in another dataset, e.g. the Human Connectome Project Data (which also has cognitive measures and pre-processed structural and diffusion MR data). Otherwise, unfortunately as it stands my opinion is that this work would be better suited to a more specialized journal. I am

unconvinced that the identified 'neuromarkers' will be clinically useful or inform us about brain function and cognitive ability.

Thank you for the suggestion. In response to the reviewer, we have added a third data cohort for validation, i.e., COBRE (Center for Biomedical Research Excellence) data, including 88 subjects (46 SZ patients and 42 HCs) with all 3 MRI modalities and the cognitive measures recorded by MCCB, who were collected via COBRE project from the University of New Mexico. Unsurprisingly, the prediction model of cognitive composite trained from FBIRN data was also able to predict MCCB composite scores for COBRE cohort, please see the updated manuscript and analyses below.

Figure 7. Identified multimodal neuromarkers and its predictability on composite cognitive scores cross three cohorts. (a) Four identified modality-specific brain networks from FBIRN cohort that were used as regressors to predict individual cognitive scores. (b) Prediction of CMINDS composite scores based on linear regression of the four regressors (mean ROI values in (a)). A correlation of $r = 0.463$ was achieved between the estimated CMINDS composite scores and its true values. (c) Generalization of the CMINDS prediction model in (b) to UNM cohort (41HCs/37SZs) to predict MCCB, $r=0.231$ (d) Generalization of the CMINDS prediction model in (b) to COBRE cohort (42HCs/46SZs) to predict MCCB, $r = 0.406$. In both (c) and (d), good generalizability of the proposed prediction model was validated, where the four mean ROI values extracted by masks in (a) were used as regressors in Eq (1). The gray regions in (b-d) indicate a 95% confidence interval was achieved between the estimated MCCB composite scores and its true values.

“The multimodal MRI data in COBRE (Center for Biomedical Research Excellence) study were preprocessed using the same pipeline for FBIRN and UNM cohorts. Based on the same linear regression model (Equation 1) trained by FBIRN and the four extracted multimodal biomarkers (GM SAN, FA CC, fALFF PFC, and fALFF pDMN, as shown in Fig. 7a), we calculated the predicted MCCB scores for both UNM and COBRE cohorts. As shown in Fig. 7d and c, Pearson

correlation of $r = 0.406$ and $r=0.236$ were achieved between the estimated MCCB composite scores and its true values for COBRE (42HCs/46SZs) and UNM (41HCs/37SZs) respectively, suggesting good generalizability of the proposed cognition-prediction model. Note that the prediction models in Fig. 7b-d are the same, namely, Eq(1), which was trained in FBIRN cohort to predict CMINDS composite.

$$\text{CMINDS composite} = -0.8 + \text{GM_SAN} \times 0.34 + \text{FA_CC} \times 0.19 + \text{fALFF_PFC} \times 0.12 + \text{ALFF_pDMN} \times 0.13 \quad (1)$$

Furthermore, we also revised Fig. 9 on the corresponding flowchart of the cognition-directed multimodal fusion and prediction analyses, by adding the third COBRE cohort prediction panel.

Figure 9. Flowchart of our cognition-directed multimodal fusion and prediction analyses. First, cognitive scores of (a) CMINDS composite score and (b) MCCB composite score were set as the reference to guide the three-way MRI fusion for the discovery cohort and replication cohort respectively. (c) To test the similarity of the identified neuromarker between cohorts, we compared the permuted spatial correlation between brain maps, and summarized the most affected cognitive domains related to schizophrenic deficit. (d) Furthermore, cognitive domain scores of attention, working memory, and verbal learning were used as reference to guide the three-way MRI fusion respectively, aiming to identify the domain-specific neuromarkers. (e) Finally, after extracting the neuromarker maps across multiple domains, we built multiple linear regression models to predict individualized cognitive scores of FBIRN cohort. The achieved models were further successfully generalized to predict corresponding cognitive measures in two independent cohorts (UNM and COBRE). The resulted figures of each step are listed below.

COBRE Subjects

“42 patients with schizophrenia and 46 age and gender matched healthy controls were included in the data set released from the Center for Biomedical Research Excellence (COBRE), University of New Mexico. All of the control participants were free of the DSM-IV diagnoses of schizophrenia and

other mental disorders. None of all participants had neurological diseases, a history of any substance dependence, or a history of clinically significant head trauma. Informed consent was obtained from all subjects according to institutional guidelines required by the Institutional Review Board. Subjects were paid for their participation. The COBRE cohort also includes the MCCB cognitive battery.”

Supplementary Table 13. Demographics and the MCCB scores of COBRE subjects

Measure	HC	SZ	p	r
Number	42	46		
Age	40.0±11.8	39.3±13.2	0.375	0.118
Gender	10F/32M	11F/35M	0.991	-0.019
MCCB				
Composite	50.8±8.7	31.3±14.6	1.7E-10	1
Speed of processing	53.6±9.0	33.3±11.8	4.0E-14	0.865
Attention/vigilance	50.2±10.0	36.3±13.5	5.3E-07	0.852
Working memory	50.3±9.8	39.6±13.6	4.9E-05	0.820
Verbal learning	45.4±8.4	37.6±8.4	4.4E-05	0.722
Visual learning	46.4±10.2	36.6±12.4	1.3E-04	0.719
Reasoning/problem solving	57.2±7.3	44.0±11.9	4.0E-08	0.806
Social cognition	51.5±10.6	42.3±12.5	3.0E-04	0.598

We have revised Fig. 7, Fig. 9, Supplementary Fig. 7, Table 1, added Supplementary Table 13, as well as rewrote the corresponding contents in the revised manuscript and Supplementary files, see page 17-20, 27 and Supplementary file page 10.

Other points–

3.3 In the null analysis where cognitive scores were permuted before the supervised fusion analysis, it is not clear why the authors chose to report voxels that appear in more than 60% of the maps, rather than show the actual maps that came out of this analysis. The useful comparison would be to see an example spatial map, e.g. from one of those permutations. Do any of the permuted maps look like the reference maps obtained using the cognitive scores? Do any of the permuted maps look like biologically plausible networks? If so, then to my mind that would indicate serious problems for the analysis.

In order to obtain the null model, we permuted the reference vector (cognitive scores) 1000 times and then used it as reference in the supervised fusion analysis. Thus we obtain 1000 three-way covarying patterns associated with the permuted reference. Therefore, it is not representative if we only show one example out of

Supplementary Figure 6. Box plot of the spatial correlations *r* between the original patterns and 1000 times permutation patterns for GM, FA and fALFF.

1000 times.

In response to the reviewer, we calculated the spatial similarity between the original pattern and the 1000 obtained maps for each modality, as displayed in Supplementary Fig. 6. Note that the highest spatial correlation is $r=0.024$, 0.015 , 0.005 for GM, FA and fALFF respectively and the mean spatial correlation is $r=0.0117$, 0.0077 , 0.0026 for GM, FA and fALFF respectively. Therefore, the permuted maps are not at all similar to the reference maps obtained using the cognitive scores.

Moreover, we can also report voxels that appear with other occurring frequency, e.g., $>40\%$, instead of $>60\%$ of the maps. In response to the reviewer, Supplementary Fig. 5 show the (a) original reference maps, (b) null patterns with voxels with $>60\%$ occurring frequency, (c) null patterns with voxels with $>40\%$ occurring frequency, and (d) one random patterns among the 1000 permutations. It is clear that the other 3 patterns in (c-d) are significantly different from the reference patterns. Only when the permuted reference is in the exact order as we used in the original analysis, are the spatial maps the same as in our reference patterns.

Supplementary Figure 5. (a) The covarying reference pattern using the original cognitive scores. (b) Null patterns with voxels with $>60\%$ occurring frequency. (c) Null patterns with voxels with $>40\%$ occurring frequency, and (d) one random pattern among the 1000 permutation.

3.4 Also - why does supervised fusion using the permuted scores give us a near perfect delineation of the lateral ventricles? Something odd is happening here, highlighting that inserting junk into the pipeline seems to produce seemingly interesting spatial patterns.

Please see response of point **#3.3**. We also noticed that for the null model, the most frequently occurring voxels happen to be within CSF and white matter, which are mainly artifacts for each modality. Since the reference used in permutation is not meaningful, we do not think there is any

linkage with the identified cognition-associated biomarkers.

3.5 I am concerned that the permutation analysis used to calculate the significance of the cross-cohort spatial correlation does not provide an adequate null model for the spatial correlations. As I've understood it, the ICref maps were calculated from smooth data which has some degree of spatial autocorrelation between neighboring voxels - due in part to neural signal, but also to non-neural factors (e.g. Kriegeskorte et al., 2008). If the voxels in the ICref map are randomly shuffled, this obliterates all spatial relationships, including these artefactual ones due to spatial proximity. In other words, the permutation null model has no proximity-related correlations whereas the cross-cohort correlations do, and comparing the two is not valid. The problem here is that the autocorrelations will inflate the cross-cohort correlation, even if the ICref map contained no biologically meaningful spatial patterns.

Artifactual time-course correlations in echo-planar fMRI with implications for studies of brain function. Kriegeskorte, N., Bodurka, J., & Bandettini, P. (2008) *International Journal of Imaging Systems and Technology*, 18, 345-349.

Cluster failure: Why fMRI inferences for spatial extent have inflated false-positive rates. Eklund, A., Nichols, T.E., & Knutsson, H. (2016) *PNAS* 113, 7900-7905.

This is a good point and we are sorry for the lack of clarity. To evaluate their relationship, we performed the random permutation procedure suggested in the paper mentioned by the reviewer "Artifactual time-course correlations in echo-planar fMRI with implications for studies of brain function. Kriegeskorte, N., Bodurka, J., & Bandettini, P. (2008) *International Journal of Imaging Systems and Technology*, 18, 345-349". Our comparison included the small subset of voxels included in both cohorts which were above a threshold. To evaluate their similarity, we performed a permutation test to calculate the significance for the cross-cohort spatial correlation on the reduced voxels. This involved randomly shuffling Y (UNM_IC_{ref}) across voxels ($m=1936$ for GM) and re-running the correlation analyses (between X [FBIRN_IC_{ref}] and Y) 10000 times in order to obtain an empirical null distribution. Take GM component for example (Fig. 3), the observed correlation between FBIRN_IC_{ref} and UNM_IC_{ref} was 0.51, while 8 of the 10000 permutations obtained correlations falling out the range of $[-0.51, 0.51]$, thus the probability of $p=8.0 \times 10^{-4}$ was estimated for cross-cohort correlation of $r=0.51$ between GM maps by chance. Importantly, the permutation testing here is only used to determine the significance (p value) of the corresponding correlation, and does not change the correlation value r .

As suggested by the reviewer, we also performed an additional permutation test by taking the local spatial relationships into consideration to avoid autocorrelation within neighboring voxels. Namely, we used a brain atlas to categorize brain voxels in component into different clusters, which

were permuted to calculate the significance further. We tested two kinds of brain atlas for fALFF component and GM component and adopted JHU atlas (49 tracts) to segment FA component. Take GM component as an example, the effective GM voxels were classified into 87 clusters based on the Brainnetome Atlas¹(total 246 ROIs). By randomly shuffling Y (UNM_IC_{ref}) across 87 clusters and re-running its correlation with X (FBIRN_IC_{ref}) for 10000 times, 2 of the 10000 permutations fell out the range of [-0.51, 0.51], thus the probability of $p=2.0 \times 10^{-4}$ was estimated for cross-cohort correlation of $r=0.51$ for GM maps. Similarly, when using AAL atlas, 46 clusters were obtained for GM component, resulting in permuted $p=4.0 \times 10^{-4}$. For FA component, 24 clusters were obtained by using JHU atlas (49 tracts), with the permuted probability of $p=5.0 \times 10^{-4}$.

Finally, we also calculated the cross-cohort correlation by correlating the mean values of 246 clusters (Brainnetome Atlas) for GM and fALFF, and 49 clusters (JHU Atlas) for FA. The resulting correlations were $r=0.67$ ($p=8.1 \times 10^{-7}$), 0.73 ($p=2.7 \times 10^{-9}$) and 0.45 ($p=1.1 \times 10^{-5}$) for GM, FA and fALFF brain maps respectively.

In sum, all three of the above approaches, designed to address spatial autocorrelation in different ways, provided similar results.

Here, in response to the reviewer, we also added another measure (Dice index, equation (2)) to calculate the overlap percentage of the spatial maps between UNM and FBIRN cohorts. Dice index is a statistical validation for comparing the spatial similarity of binary images, for example in image segmentation accuracy assessment. We calculated the dice index of the identified target component between two cohorts using only voxels masked at $|Z|>2$ (threshold), resulting in two masks from FBIRN (mask_FBIRN) and UNM (mask_UNM) respectively. Only voxels that fell in the union of the masks (mask_FBIRN \cup mask_UNM) were used to calculate the cross-cohort similarity as shown in equation (2).

$$\text{Dice index} = 2 \frac{V(A \cap B)}{V(A) + V(B)} \quad (2)$$

The Dice index for GM, FA and fALFF components are 0.75, 0.78 and 0.62 respectively, suggesting that there is high overlap percentage of the spatial maps cross FBIRN and UNM. Furthermore, with respect to the reviewer mentioned ‘‘Cluster failure: Why fMRI inferences for spatial extent have inflated false-positive rates. Eklund, A., Nichols, T.E., & Knutsson, H. (2016) PNAS 113, 7900-7905’’, we did not perform statistical analyses using SPM, AFNI or FSL. SPM is only used for fMRI and sMRI preprocessing, resulting in whole-brain voxel-wise features for further supervised fusion analysis. Thus we do not believe the false-positive results highlighted in the cluster analysis based on SPM, AFNI and FSL is present in our analysis.

Combining all the above evidence together, we are confident that the cross-cohort spatial similarity is consistent high.

3.6 I appreciate the effort to clarify the terms in the manuscript. Some aspects are still confusing though, for instance, the "multimodal neuromarker network" seems to be comprised of three networks, the DMN, SAN and CEN. Is there not a separate term that the authors could use for the identified marker than the word "network"? Such as, a 'signature' or just a 'multimodal marker' or 'imaging correlate' etc?

Thank you for the suggestion. We have changed the “multimodal neuromarker network” to “multimodal neuromarker signature” throughout the paper.

3.7 Similarly, the authors have a very specific definition of the term 'neuromarker': "Here a "neuromarker" is defined as a brain measure that is associated with a cognitive or behavioral outcome and can further predict individual performance". If the authors don't want to define "neuromarker" in the abstract, they should use a different term instead. For instance, the term "biomarker" is commonly used in the field. The first sentence of abstract doesn't make sense without the term 'neuromarker' being defined.

Thank you for the suggestion. We have replaced the term “neuromarker” in the abstract with “biomarker”, see page No. 2.

1. Fan L, *et al.* The Human Brainnetome Atlas: A New Brain Atlas Based on Connectional Architecture. *Cerebral cortex* **26**, 3508-3526 (2016).

REVIEWERS' COMMENTS:

Reviewer #2 (Remarks to the Author):

The authors have answered all my concerns

Reviewer #3 (Remarks to the Author):

The authors have addressed my concerns.

Reviewer #2 (Remarks to the Author):

2.1 The authors have answered all my concerns.

Thank you for your comments.

Reviewer #3 (Remarks to the Author):

3.1 The authors have addressed my concerns.

Thank you for your comments.